# CARD9 negatively regulates NLRP3-induced IL-1β production on Salmonella infection of macrophages

Milton Pereira[1], Panagiotis Tourlomousis[1], John Wright[1], Tom P. Monie[2] & Clare E. Bryant[1]

Interleukin-1β (IL-1β) is a proinflammatory cytokine required for host control of bacterial infections, and its production must be tightly regulated to prevent excessive inflammation. Here we show that caspase recruitment domain-containing protein 9 (CARD9), a protein associated with induction of proinflammatory cytokines by fungi, has a negative role on IL-1β production during bacterial infection. Specifically, in response to activation of the nucleotide oligomerization domain receptor pyrin-domain containing protein 3 (NLRP3) by Salmonella infection, CARD9 negatively regulates IL-1β by fine-tuning pro-IL-1β expression, spleen tyrosine kinase (SYK)-mediated NLRP3 activation and repressing inflammasome-associated caspase-8 activity. CARD9 is suppressed during *Salmonella enterica* serovar Typhimurium infection, facilitating increased IL-1β production. CARD9 is, therefore, a central signalling hub that coordinates a pathogen-specific host inflammatory response.

[1] Department of Veterinary Medicine, University of Cambridge, Madingley Road, Cambridge CB3 0ES, UK. [2] Medical Research Council Human Nutrition Research, Elsie Widdowson Laboratory, 120 Fulbourn Road, Cambridge CB1 9NL, UK. Correspondence and requests for materials should be addressed to T.M. (email: thomas.monie@mrc-hnr.cam.ac.uk) or to C.E.B. (email: ceb27@cam.ac.uk).

nterleukin-1β (IL-1β) is a cytokine of critical importance in inflammatory, infectious and autoimmune diseases. Its powerful proinflammatory effects mean that production of this cytokine must be tightly regulated to prevent excessive inflammation. Pro-IL-1β is cleaved to bioactive IL-1β by several enzymatic complexes, the most important of which are the inflammasomes. Inflammasomes assemble in the cytoplasm of cells and utilize caspases, such as caspase-1 and caspase-8, for cytokine processing[1]. Canonical inflammasomes comprise a nucleotide oligomerization domain receptor (NLR), an adaptor protein (ASC; apoptosis-associated speck-like protein containing a CARD (caspase recruitment domain)) and effector caspases. Non-canonical inflammasomes are formed in response to activation of other pattern recognition receptors (PRRs), such as the complex formed by dectin-1, spleen tyrosine kinase (SYK) and caspase-8 in response to fungal infection[2–4].

Formation of the macromolecular inflammasome structure requires protein–protein interactions, often between CARD domain-containing proteins, for example, between ASC and caspase-1, NLR family CARD domain-containing protein 4 (NLRC4) and ASC or NLRC4 and caspase-1. Canonical inflammasomes can recruit multiple NLRs and caspases to the same inflammasome complex to tailor a pathogen-specific inflammatory response[5]. This recruitment occurs through multiple domain-specific protein interactions, including those mediated by CARDs. Identifying the final composition of inflammasomes formed in response to specific pathogens or cellular insults, and how this affects inflammasome activity, is likely to be of major clinical importance in infectious and inflammatory disease research.

CARD9 is a CARD-containing adaptor protein with roles in activating innate immune signalling, particularly in response to fungal[6] and viral infections[7,8]. CARD9 regulates dectin-1 stimulation of nuclear factor-κB (NF-κB) activation[9], reactive oxygen species production[10] and non-canonical inflammasome assembly[3,4], as well as activation of p38 and Jnk during nucleotide-binding oligomerization domain-containing protein 2 (NOD2) signalling[11]. Mutations in CARD9 are associated with chronic inflammatory diseases in humans, including those associated with bacterial rather than fungal infection, such as Crohn's disease and colitis[12–16]. Little is known about whether CARD9 is important in regulating the host response to bacterial infection. CARD9 upregulates IL-1β production in fungal infections, but whether there is a direct link between this protein and canonical inflammasome activity is unclear.

Infection with Salmonella enterica serovar Typhimurium (S. Typhimurium) triggers the formation of a complex inflammasome that can include NLRC4, NLR family PYRIN domain-containing protein 3 (NLRP3), ASC, caspase-1 and caspase-8 (refs 5,17–19). Caspase-1 drives IL-18 and IL-1β maturation and gasdermin D processing, which leads to pyroptosis[20,21]. Caspase-8 is important for IL-1β maturation but has no clear role in pyroptosis[17]. The complex composition of the Salmonella-induced inflammasome led us to speculate whether other CARD domain-containing proteins, such as CARD9, may regulate its activity. CARD9 regulates SYK activity and this kinase phosphorylates the CARD domain of ASC when NLRP3, but not NLRC4, is activated to increase IL-1β and IL-18 production[22–25]. We investigated whether CARD9 could regulate canonical inflammasome activation in response to S. Typhimurium infection. Surprisingly, we show that in response to Salmonella infection CARD9 negatively regulates NLRP3-induced IL-1β production, but not pyroptosis, in murine bone marrow-derived macrophages (BMDMs) by two distinct mechanisms: (i) by fine-tuning pro-IL-1β expression; and (ii) by reducing NLRP3 activation through modulation of SYK and caspase-8 activity. Importantly, we show that CARD9 can negatively regulate canonical NLRP3 inflammasome activity and NOD2-mediated pro-IL-1β synthesis. Our data identify a negative regulatory role for CARD9 on IL-1β production in bacterial infection that contrasts its role in fungal infections in which it drives proinflammatory responses. We propose that CARD9 is a central signalling hub that can negatively or positively regulate proinflammatory signalling to coordinate a pathogen-specific host inflammatory response to infection.

## Results

**CARD9 negatively regulates IL-1β from inflammasomes.** To evaluate whether CARD9 plays a role in inflammasome activity, we infected BMDMs from wild type (WT), $Card9^{-/-}$ and $Nlrc4^{-/-}$ C57BL/6 mice with S. Typhimurium SL1344. As expected[26], $Nlrc4^{-/-}$ BMDMs had reduced IL-1β processing and induction of cell death at all multiplicities of infection (m.o.i.'s) at early time points, consistent with impaired inflammasome assembly (Fig. 1a,b,d,e,g,h). $Card9^{-/-}$ BMDMs, however, showed a slight reduction in S. Typhimurium-induced cell death. Decreased pyroptosis is usually coupled to reduced IL-1β secretion[1], but here we observed the opposite effect in $Card9^{-/-}$ BMDMs. At all time points and m.o.i.'s studied, there was an increase in IL-1β in the supernatant from $Card9^{-/-}$ BMDMs of up to four times that seen from WT cells (Fig. 1a,d,g). The increased cytokine production could not be explained by increased intracellular bacterial load, as both WT and $Card9^{-/-}$ BMDMs had comparable intracellular counts (Supplementary Fig. 1a–c). We measured tumour-necrosis factor-α (TNF-α) production from the same cell supernatant and showed that although there is an increase in the production of this cytokine the amount produced is proportional to the increase in cell viability, whereas the production of IL-1β is disproportionally enhanced from the infected $Card9^{-/-}$ cells (Fig. 1a,c,d,f,g,i). BMDM infection with Escherichia coli also showed increased IL-1β production in $Card9^{-/-}$ in comparison with WT macrophages without affecting pyroptosis, intracellular bacteria counts and TNF-α production (Fig. 1j–o and Supplementary Fig. 1d–e). It is possible that the effects of CARD9 on Salmonella-induced inflammasome activity may be indirect, for example, by regulating the autocrine production of cytokines such as IL-10, which is reduced in CARD9-deficient neutrophils infected with Mycobacterium tuberculosis[27]. An IL-10-mediated effect of CARD9 on Salmonella-induced IL-1β production from macrophages is unlikely, however, as elevated IL-1β production occurs within 2 h of infection, whereas IL-10 production occurs later in infection[28]. The increased IL-1β production in vitro is also seen in the production of this cytokine from a sub-lethal in vivo model of systemic infection with S. Typhimurium. WT and $Card9^{-/-}$ mice infected with S. Typhimurium (strain M525P) had similar bacterial burdens in the spleen at days 1, 3 and 7 post infection (Supplementary Fig. 1f–g). Immunoblotting of cell lysates from isolated spleen cells revealed an upregulation of pro-IL-1β in $Card9^{-/-}$ mice at day 7, without an increase in the caspase-1 p10 subunit (Fig. 1p,q). The mature cytokine could not be reliably measured in mouse sera at this time point, consistent with other published data[26].

One explanation for these data could be the well-known role of CARD9, during fungal infection, in regulating NF-κB signalling pathways[6] such that enhanced expression of pro-IL-1β could result in an overall increase in IL-1β production without effecting inflammasome-induced cytokine processing and pyroptosis. BMDMs were infected with S. Typhimurium at an m.o.i. 5 for 2 h and, following mRNA extraction, we performed quantitative

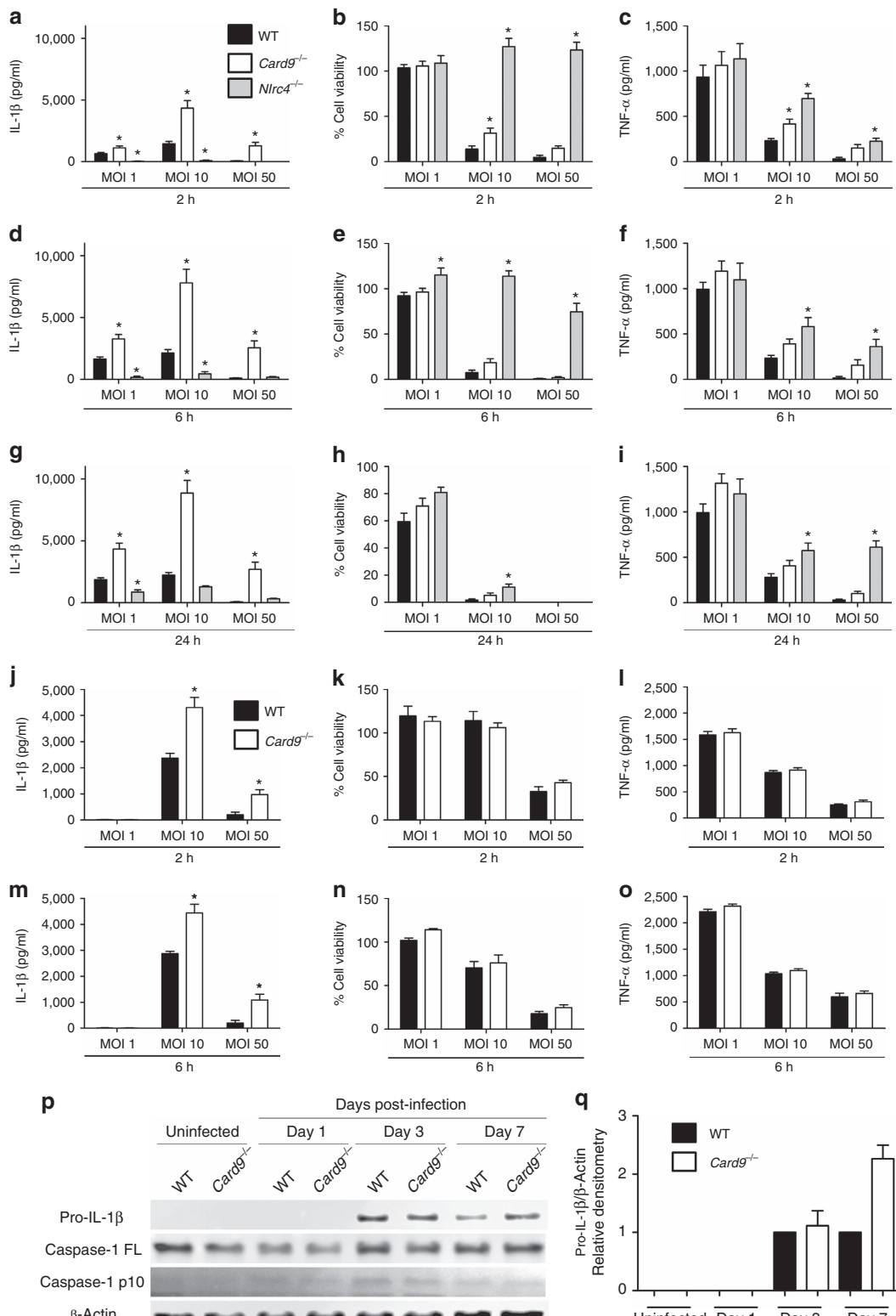

**Figure 1 | CARD9 negatively regulates IL-1β production *in vitro* and *in vivo*.** (**a**,**d**,**g**) IL-1β secretion (as measured by ELISA), (**b**,**e**,**h**) cellular viability (as measured by LDH release) and (**c**,**f**,**i**) TNF-α (as measured by ELISA) of WT, $Nlrc4^{-/-}$ and $Card9^{-/-}$ BMDMs after infection with *S.* Typhimurium SL1344 at m.o.i.'s 1, 10 and 50 for 2 (**a–c**), 6 (**d–f**) and 24 (**g–i**) h. (**j**,**m**) IL-1β secretion, (**k**,**n**) cellular viability and (**l**,**o**) TNF-α after infection of WT and $Card9^{-/-}$ BMDMs with *E. coli* P19A at m.o.i.'s 1, 10 and 50 for 2 (**j–l**) and 6 (**m–o**) h. (**p**) Immunoblot analysis of pro-IL-1β, caspase-1 and β-actin in spleen cells isolated from infected WT and $Card9^{-/-}$ C57BL/6 mice after intravenous infection with *S.* Typhimurium M525P ($4 \times 10^3$ colony-forming units). (**q**) Densitometric analysis of this immunoblot. *$P < 0.05$ (one-way analysis of variance with Tukey's multiple comparisons test). (**a–i**) Data from five independent experiments (mean and s.e.m.). (**j–o,q**) Data from two independent experiments (mean and s.e.m). (**p**) Representative data from two independent experiments, using cells pooled from four to six mice per genotype, plus two negative controls per genotype. Mice were from 8 to 16 weeks old, both male and female.

PCR analysis for genes involved in inflammasome signalling, NF-κB target genes and other genes coding for proteins known to associate with CARD9. Most NF-κB target genes, such as TNF-α and RANTES, were similarly upregulated in WT and $Card9^{-/-}$ BMDMs (Fig. 2c,d). Levels of pro-IL-1β, however, whilst significantly upregulated in WT cells (around 800-fold increase) were even more enhanced in $Card9^{-/-}$ cells (1,200 fold-increase, $P < 0.01$; Fig. 2a). This pattern of enhanced pro-IL-1β was also observed at the protein level (Fig. 2g). NLRP3 was upregulated to a similar level in both WT and $Card9^{-/-}$ BMDMs (Fig. 2b). SYK and caspase-8, proteins know to be involved in non-canonical inflammasome activity and NLRP3 activation[18,24], were also found to be slightly upregulated after

infection of $Card9^{-/-}$ BMDMs at both the mRNA and protein level (Fig. 2e–i). Several other genes (Naip5, Nlrc4, Pycard, Casp1, Fadd, Bcl10, Malt1 and Il18) showed no difference in expression between WT and $Card9^{-/-}$-infected BMDMs (Supplementary Fig. 2a–h). None of the analysed transcripts or proteins showed basal differences between uninfected WT and $Card9^{-/-}$ cells (Supplementary Fig. 2i–j).

To determine whether CARD9 affects inflammasome activation, as well as pro-IL-1β expression we analysed, by immunoblotting, the BMDM supernatants for pro-IL-1β and caspase-1 processing. Our data show increased IL-1β protein expression in $Card9^{-/-}$ BMDMs at 2 and 6 h post infection compared with WT cells. As expected BMDMs from ASC

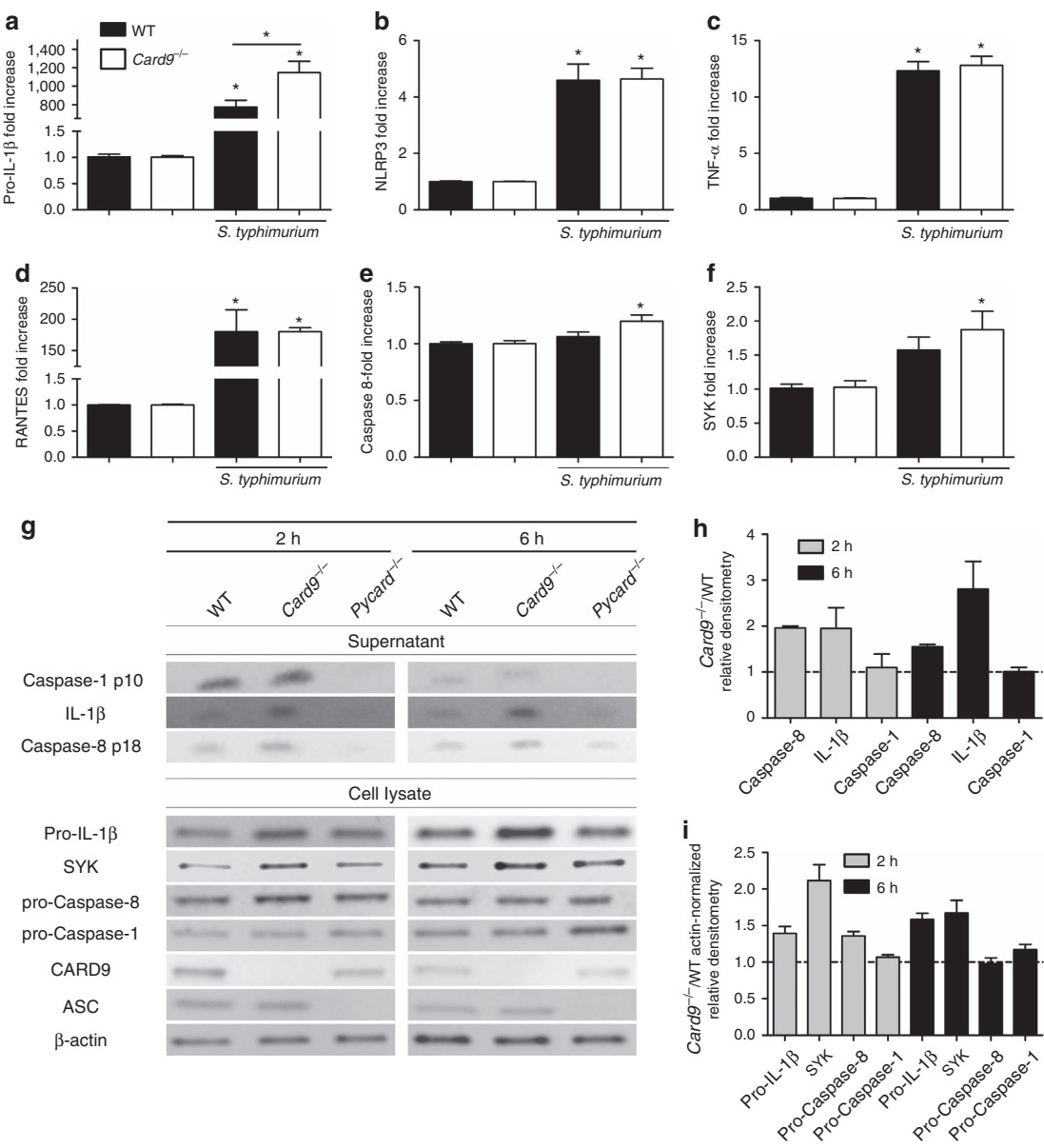

**Figure 2 | CARD9 modulates pro-IL-1β expression.** (**a–f**) Quantitative PCR analysis of WT and $Card9^{-/-}$ BMDMs infected with S. Typhimurium (m.o.i. 5) for 2 h, compared with uninfected controls. (**a**) Pro-IL-1β, (**b**) NLRP3, (**c**) TNF-α, (**d**) RANTES, (**e**) caspase-8 and (**f**) SYK. (**g**) Expression of pro-IL-1β, SYK, pro-caspase-8, pro-caspase-1, CARD9, ASC, β-actin in cell lysates, and caspase-1 p10, caspase-8 p18 and IL-1β in culture supernatants from WT, $Card9^{-/-}$ and $Pycard^{-/-}$ BMDMs after 2 or 6 h of infection with S. Typhimurium (m.o.i. 5) and relative densitometry ($Card9^{-/-}$/WT) in culture supernatants (**h**) or cell lysates ($Card9^{-/-}$/WT, actin-normalized) (**i**). *$P < 0.05$ in comparison with uninfected control, unless stated in the graph (one-way analysis of variance with Tukey's multiple comparisons test). (**a–f,h,i**) Data from three independent experiments (mean and s.e.m.). (**g**) Image is representative of three independent experiments.

knockout mice ($Pycard^{-/-}$) were impaired in the production of IL-1β and the mature forms of both caspase-1 and caspase-8. Caspase-1 processing was similar in both WT and $Card9^{-/-}$ BMDMs at 2 and 6 h, suggesting that the increased IL-1β production in $Card9^{-/-}$ cells is not due to an increase in caspase-1 activation. We recently showed that caspase-8 plays an important role in canonical inflammasome processing of IL-1β in response to S. Typhimurium, but not pyroptosis, which is wholly dependent on caspase-1 (ref. 17) via cleavage and activation of Gasdermin D[20,21]. The mature form of caspase-8 was increased in $Card9^{-/-}$ BMDMs at 2 h (Fig. 2g,h) suggesting that, in the absence of CARD9, increased caspase-8 activity is responsible for the increased conversion of pro-IL-1β to its mature form.

**CARD9 functions independently of NLRC4 and AIM2.** S. Typhimurium stimulates IL-1β production via both NLRC4 and NLRP3 activation[26]. To assess whether CARD9 affects IL-1β production by each of these NLRs LPS-primed WT, $Nlrc4^{-/-}$, $Nlrp3^{-/-}$ and $Card9^{-/-}$ BMDMs were transfected with ultrapure flagellin from Salmonella for 1 h, followed by quantification of IL-1β production. IL-1β was elevated to similar

levels in WT, NLRP3 and CARD9 knockout macrophages, but, as expected, was below the detection limit for $Nlrc4^{-/-}$ BMDMs (Fig. 3a). CARD9 modulation of infla-'mmasome-induced IL-1β production therefore occurs independently of NLRC4.

To confirm this observation, we infected WT BMDM with either WT S. Typhimurium or a mutant strain deficient in NLRC4 activation[5] (ΔfliCΔfljBΔprgJ). Both WT and $Card9^{-/-}$ BMDM infected with the ΔfliCΔfljBΔprgJ strain had impaired levels of pyroptosis in comparison with cells infected with WT bacteria (Fig. 3b–d). IL-1β secretion was elevated in BMDMs infected with WT S. Typhimurium, but markedly reduced, as expected, in BMDMs after infection with the ΔfliCΔfljBΔprgJ mutant. In $Card9^{-/-}$ BMDMs, however, IL-1β production was increased after infecting cells with both strains of bacteria (Fig. 3e–g). At 2 h post infection $Card9^{-/-}$ BMDMs produced detectable IL-1β after infection with both bacterial strains although the level was much reduced in cells infected with the ΔfliCΔfljBΔprgJ mutant. No IL-1β was detected in WT cells at this time point infected with the ΔfliCΔfljBΔprgJ strain. $Card9^{-/-}$ macrophages secreted 2.5 times more IL-1β than WT BMDMs at both 6 and 24 h post infection, time points known to

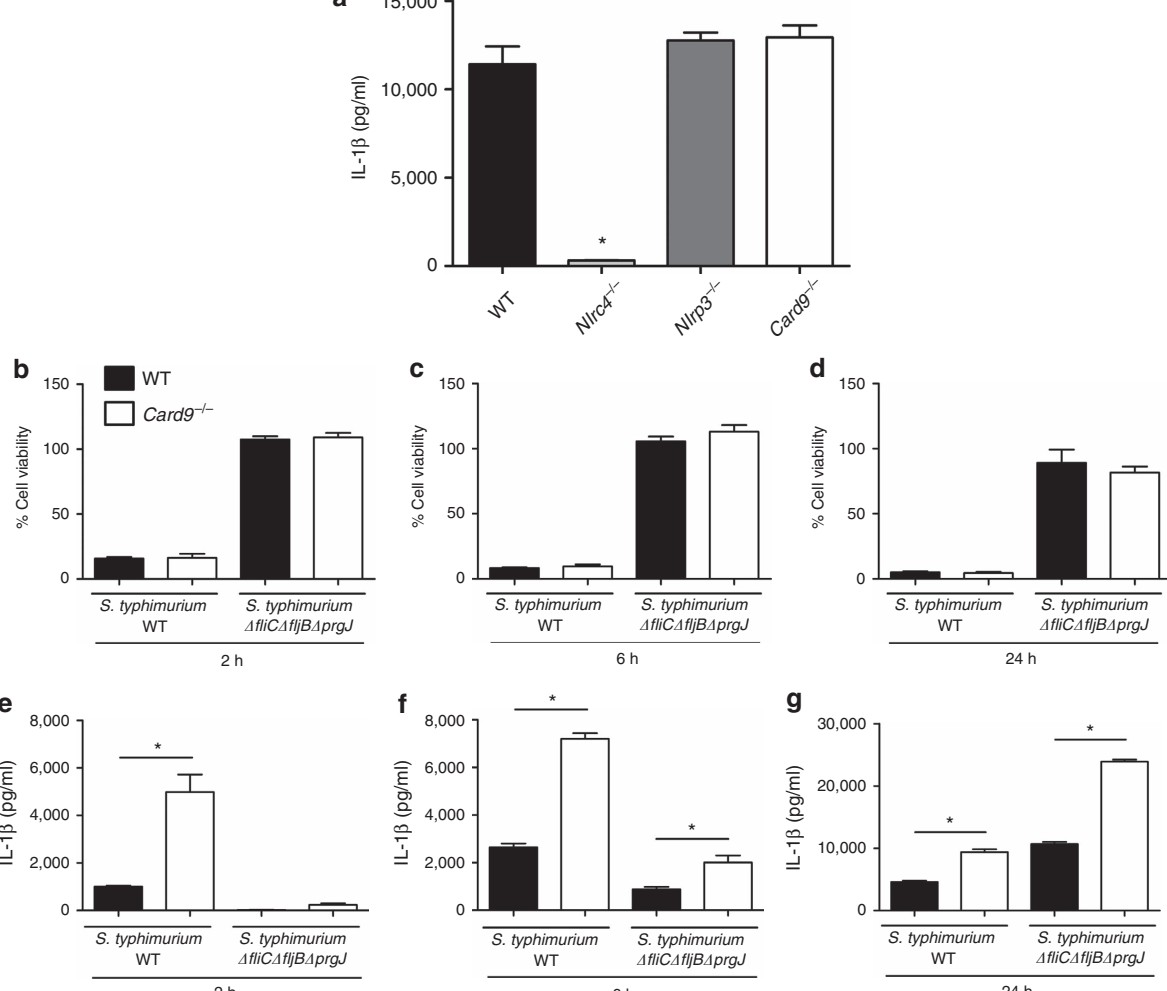

**Figure 3 | CARD9 does not control IL-1β produced via NLRC4.** (**a**) IL-1β production from LPS-primed BMDMs after transfection with ultrapure flagellin. (**b–d**) Cellular viability and (**e–g**) IL-1β production from unprimed BMDMs after infection with S. Typhimurium WT or ΔfliCΔfljBΔprgJ (deficient in NLRC4 activation) at an m.o.i. 10 for 2 (**b,e**), 6 (**c,f**) and 24 (**d,g**) h. *$P < 0.05$ in comparison with WT (one-way analysis of variance with Tukey's multiple comparisons test). (**a–g**) Data from three independent experiments (mean and s.e.m.).

involve NLRP3 activity in response to *Salmonella* infection[5]. This is consistent with our flagellin data suggesting that IL-1β produced via NLRC4 occurs independently of CARD9.

To determine whether AIM2 inflammasome activity could be regulated by CARD9, LPS-primed WT and *Card9*[−/−] BMDMs were stimulated with the AIM2 ligand poly(dA:dT), but no differences in cellular viability or IL-1β secretion were seen (Supplementary Fig. 3).

**CARD9 negatively regulates IL-1β production through NLRP3**. To determine whether the NLRC4-independent impact of CARD9 on IL-1β production resulted from activation of NLRP3 we infected BMDMs with *S*. Typhimurium at an m.o.i. 10 in the presence or absence of either glibenclamide or MCC950 (refs 29,30). At 2 and 6 h post infection, both WT and *Card9*[−/−] BMDMs showed similar levels of cellular viability in the presence or absence of glibenclamide (Fig. 4a,b), while MCC950 slightly inhibited cell death in *Card9*[−/−] BMDM (Fig. 4e,f). Glibenclamide and MCC950 did not affect IL-1β production in WT cells, but they did reduce the enhancement of IL-1β production in *Card9*[−/−] BMDMs to a level comparable to that seen in WT cells. These data suggest that the increased IL-1β production from *Card9*[−/−] macrophages after *Salmonella* infection is driven by enhanced NLRP3 activation (Fig. 4c,d,g,h).

To verify whether CARD9 regulates IL-1β production via NLRP3, LPS-primed (200 ng ml[−1] for 3 h) WT, *Nlrc4*[−/−], *Nlrp3*[−/−] and *Card9*[−/−] BMDMs were stimulated for 1 h with the NLRP3 activator nigericin (10 μM, 1 h). Nigericin, as expected, stimulated cell death and IL-1β secretion in WT and *Nlrc4*[−/−], but not in *Nlrp3*[−/−], BMDMs (Fig. 4i,j). Levels

of IL-1β produced by *Card9*[−/−] BMDMs were greatly enhanced in comparison with WT cells after stimulation with nigericin. Similar levels of cellular viability were seen in nigericin-stimulated WT and *Card9*[−/−] BMDMs (Fig. 4i,j). Similarly, incubation of WT, *Card9*[−/−] and *Nlrp3*[−/−] BMDMs with ATP showed no alterations in cellular viability (Fig. 4k) and a significant increase in IL-1β production in *Card9*[−/−] cells in comparison with WT cells, while *Nlrp3*[−/−] BMDMs showed great impairment in IL-1β production (Fig. 4l). These data confirm that the effects of CARD9 on inflammasome function are specific for NLRP3-mediated IL-1β production and are uncoupled from inflammasome-driven pyroptosis.

**CARD9 inhibits NLRP3 activation upstream of speck formation**. The enhanced production of NLRP3-dependent IL-1β secretion in *Card9*[−/−] BMDMs is independent of caspase-1 (Fig. 2g–i). How then does the presence of CARD9 inhibit processing of IL-1β? We investigated first whether CARD9 acts upstream or downstream of the canonical inflammasome adaptor protein ASC. WT, *Card9*[−/−] and *Pycard*[−/−] LPS-primed BMDMs were stimulated with nigericin (5 μM for 30 min), fixed and stained for ASC and CARD9. Both WT and *Card9*[−/−] cells formed ASC specks (Fig. 5a). Immunolocalization of endogenous CARD9 suggests that this protein forms aggregates in stimulated and unstimulated WT and *Pycard*[−/−] BMDMs, but it does not co-localize with ASC specks suggesting that CARD9 is not recruited to the ASC speck (Fig. 5a). *Card9*[−/−] BMDMs show some non-specific background staining, but it is much fainter than the CARD9 immunolocalization in WT and *Pycard*[−/−] cells.

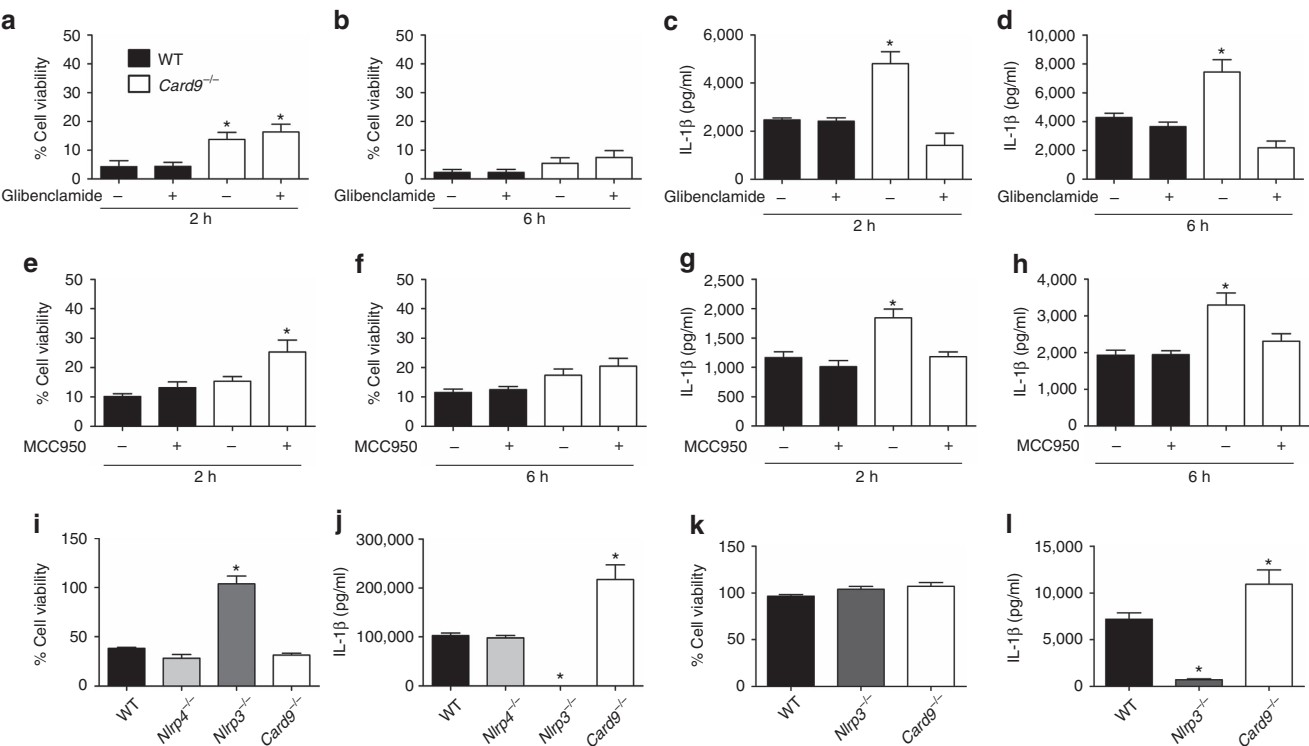

**Figure 4 | CARD9 selectively negatively regulates NLRP3-induced IL-1β production.** (**a,b,e,f**) Cellular viability and (**c,d,g,h**) IL-1β secretion from unprimed WT and *Card9*[−/−] BMDMs infected with *S*. Typhimurium at m.o.i. 10 for 2 (**a,c,g**) and 6 (**b,d,f,h**) h in presence or absence of NLRP3 inhibitors glibenclamide (**a–d**) and MCC950 (**e–h**). (**i,j**) Cellular viability (**i**) and IL-1β secretion (**j**) from LPS-primed BMDMs after nigericin stimulation (10 μM, 1 h). IL-1β from LPS-primed cells without nigericin stimulation was below the level of detection. (**k,l**) Cellular viability (**k**) and IL-1β secretion (**l**) from LPS-primed BMDMs after ATP stimulation (5 mM, 30 min). *P < 0.05 in comparison with WT (one-way analysis of variance with Tukey's multiple comparisons test). (**a–l**) Data from three independent experiments (mean and s.e.m.).

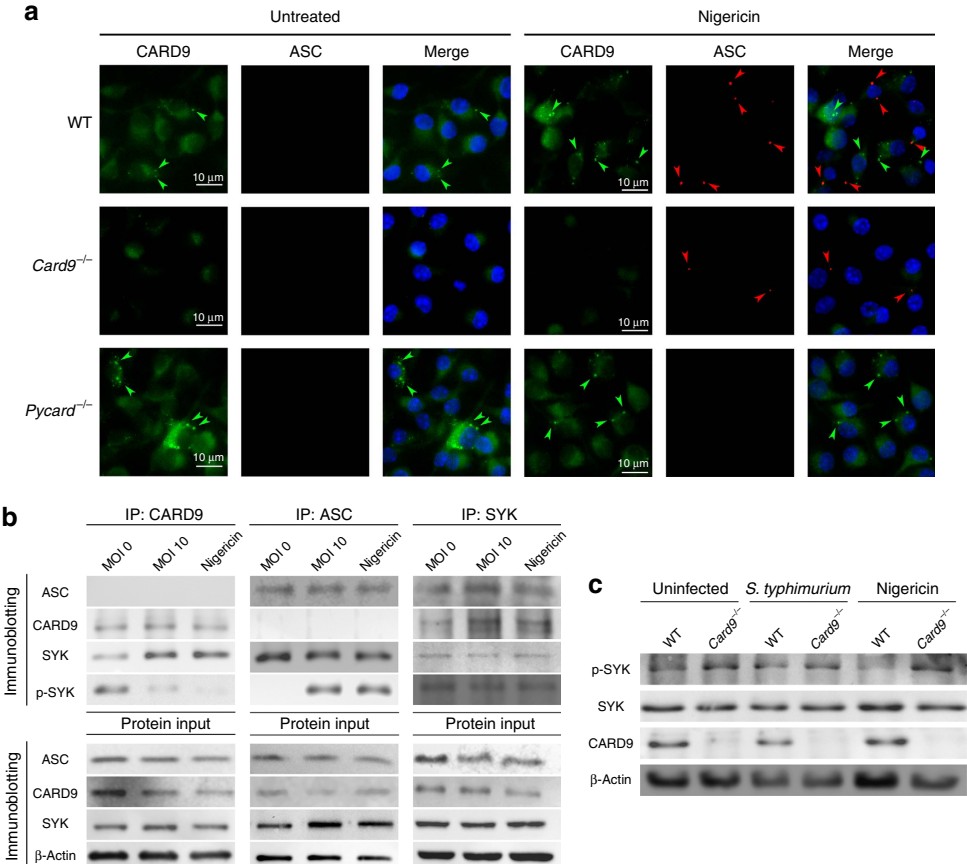

**Figure 5 | CARD9 and SYK regulates NLRP3 activation upstream to speck formation.** (**a**) CARD9 and ASC were immuno-labelled in WT, $Card9^{-/-}$ and $Pycard^{-/-}$ LPS-primed macrophages after nigericin stimulation (5 μM, 30 min). Green arrows indicate CARD9 aggregates. Red arrows indicate ASC specks. (**b**) Co-immunoprecipitation of ASC, CARD9 and SYK from cell lysates of uninfected, *S.* Typhimurium-infected (m.o.i. 10, 30 min) or nigericin-stimulated (10 μM, 30 min) LPS-primed BMDMs. (**c**) Immunoblot analysis of LPS-primed WT and $Card9^{-/-}$ uninfected, *S.* Typhimurium-infected (m.o.i. 10, 2 h) or nigericin-stimulated (10 μM, 30 min) BMDMs. (**a–c**) Images are representative of three independent experiments. Scale bar, 10 μm.

To determine whether CARD9 associates with ASC before speck formation co-immunoprecipitation analysis of WT BMDM lysate was performed using anti-ASC, anti-CARD9 or anti-SYK antibodies as bait. SYK was targeted because of its involvement in both CARD9 and ASC signalling[22,24]. Co-immunoprecipitations confirmed that both ASC and CARD9 interact with SYK in unstimulated or uninfected WT BMDMs. ASC and CARD9 do not, however, directly interact with one another (Fig. 5b). After *Salmonella* infection or nigericin treatment, ASC immunoprecipitated preferentially with the phosphorylated form of SYK (p-SYK) confirming previously published data[25], while CARD9 predominantly interacted with unphosphorylated SYK (Fig. 5b). Phosphorylated SYK regulates NLRP3 activation[22–25], so the interaction of CARD9 with unphosphorylated SYK may prevent its subsequent phosphorylation thereby inhibiting NLRP3 activation. No proteins were pulled down in $Card9^{-/-}$ and $Pycard^{-/-}$ isotype controls IPs (Supplementary Fig. 4). Consistent with this hypothesis we found increased phosphorylation of SYK in unstimulated, infected or nigericin-treated BMDM from $Card9^{-/-}$ mice in comparison with WT cells (Fig. 5c). This supports the idea that in the absence of CARD9 there is increased SYK phosphorylation, which facilitates NLRP3 activation.

**CARD9 inhibits NLRP3 activation through SYK and caspase-8.** Our immunoblotting (Fig. 2g), co-immunoprecipitation (Fig. 5b)

and SYK phosphorylation data (Fig. 5c) all suggest that CARD9 regulates NLRP3 activation by a SYK and caspase-8-dependent mechanism. To investigate this hypothesis LPS-primed BMDMs were infected in the presence or absence of R406, a specific SYK inhibitor, and Z-IETD-FMK, a caspase-8 inhibitor. The expression of pro-IL-1β, caspase-8, caspase-1, SYK and CARD9 are similar in LPS-primed WT and $Card9^{-/-}$ BMDM in the presence or absence of these inhibitors (Supplementary Fig. 5). BMDM infection with *S.* Typhimurium (m.o.i. 10) in the presence of the caspase-8 inhibitor Z-IETD-FMK, as expected, had no effect on cell viability as measured by lactate dehydrogenase (LDH) activity (Fig. 6a) because caspase-8 does not induce pyroptosis in response to infection with this pathogen[17]. It is possible, however, that caspase-8 may induce cell death in response to infection by other mechanisms[31]. IL-1β production was reduced in both WT and $Card9^{-/-}$ macrophages in the presence of Z-IETD-FMK, which is consistent with our data suggesting that caspase-8 plays an important role in IL-1β processing during *Salmonella* infection[17]. Importantly IL-1β secretion in infected $Card9^{-/-}$ macrophages, in the presence of Z-IETD-FMK, was now no different to that of infected WT BMDM (Fig. 6b) supporting the hypothesis that the increased IL-1β production observed in $Card9^{-/-}$ cells is indeed mediated by caspase-8. Treatment with Z-IETD-FMK also reduced IL-1β secretion from nigericin-stimulated $Card9^{-/-}$ BMDMs to the same level as WT cells (Fig. 6c,d).

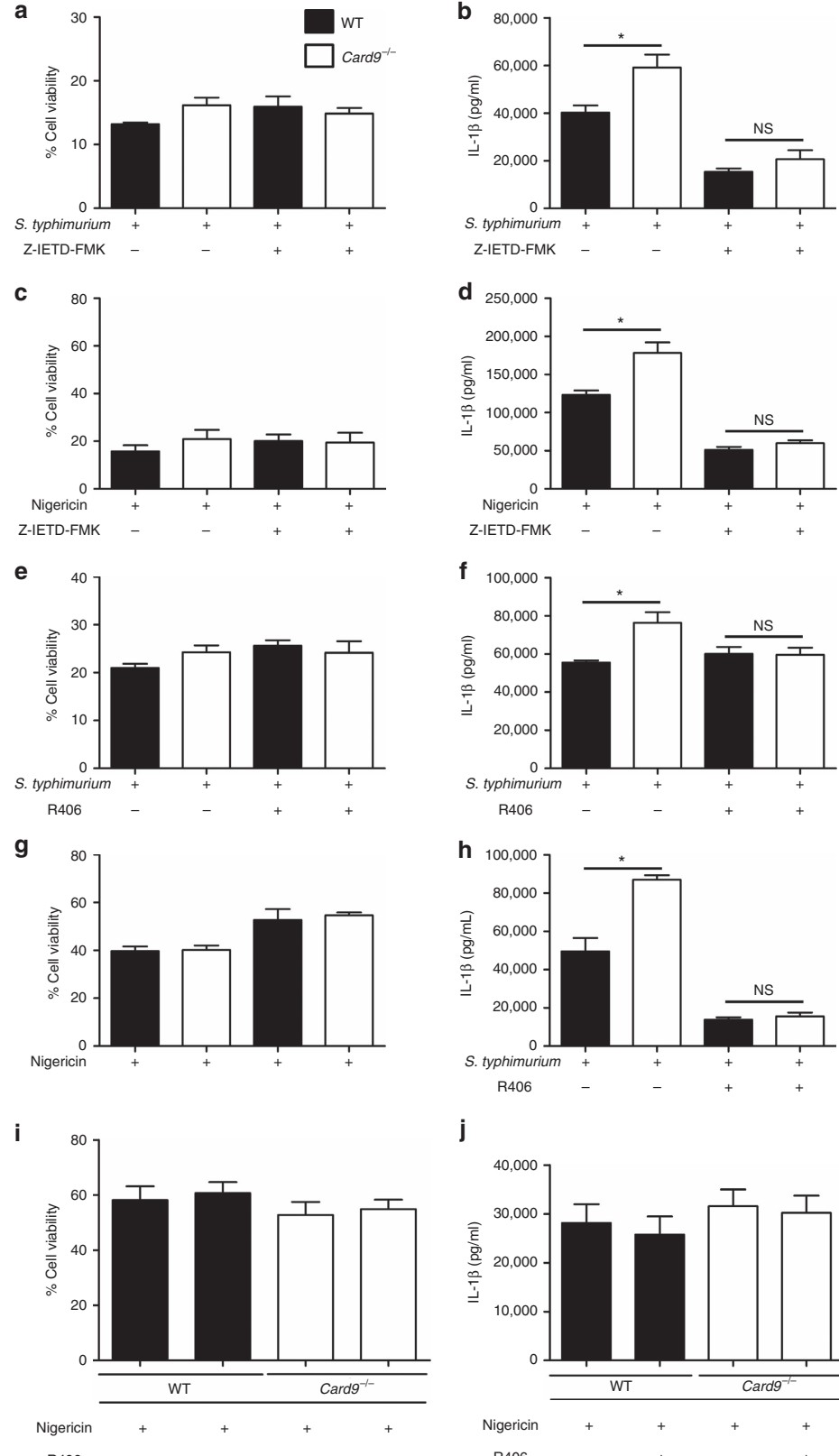

**Figure 6 | SYK and caspase-8 activity are important for CARD9 regulation of NLRP3 activity in BMDMs, but not in BMDCs.** (**a–d**) Effects of caspase-8 inhibition by Z-IETD-FMK on LPS-primed WT and $Card9^{-/-}$ LPS-primed BMDMs on cellular viability (**a,c**) and IL-1β production (**b,d**) during $S$. Typhimurium infection (m.o.i. 10, 1 h) (**a,b**) or nigericin (10 μM, 1 h) stimulation (**c,d**). (**e–h**) Effects of SYK inhibition by R406 on WT and $Card9^{-/-}$ LPS-primed BMDMs on cellular viability (**e,g**) and IL-1β secretion (**f,h**) in response to $S$. Typhimurium infection (m.o.i. 10, 1 h) (**e,f**) or nigericin stimulation (5 μM, 1 h) (**g,h**). (**i,j**) Stimulation of WT or $Card9^{-/-}$ BMDCs with nigericin (5 μM, 1 h). Cellular viability (**i**) and IL-1β secretion (**j**). *$P < 0.05$ (one-way analysis of variance with Tukey's multiple comparisons test. (**a–j**) Data from three independent experiments (mean and s.e.m.).

Cell viability in BMDMs infected with *S.* Typhimurium in the presence of the SYK inhibitor R406 was unaffected, as expected, given that SYK activity does not regulate NLRC4 activity (Fig. 6e)[24]. SYK inhibition decreased the amount of IL-1β secreted by infected *Card9*[−/−] BMDMs to similar levels to those seen in WT cells (Fig. 6f). SYK inhibition, however, had less effect on IL-1β production than caspase-8 inhibition probably because caspase-8 is activated by both NLRC4 and NLRP3 in response to *Salmonella* infection[17], whereas SYK only regulates NLRP3 activation[24]. Stimulation of LPS-primed WT or *Card9*[−/−] BMDMs with nigericin in the presence of R406 elicited similar levels of IL-1β production from both cell types (Fig. 6h). SYK inhibition led to a small, but statistically significant, increase in cell viability in both WT and CARD9 knockout BMDM stimulated with nigericin probably due to its effect on NLRP3-induced caspase-1 activation[25] (Fig. 6g).

SYK regulates NLRP3 activity in BMDM, but it does not affect NLRP3 activation in bone marrow-derived dendritic cells (BMDCs)[24]. If CARD9 plays a specific role in regulating NLRP3 activity through SYK then there should be no difference in NLRP3-stimulated IL-1β production between WT and *Card9*[−/−] BMDC. Stimulation of LPS-primed BMDCs with nigericin (5 μM, 1 h) induced similar levels of IL-1β processing and cell death in both *Card9*[−/−] and WT BMDCs (Fig. 6i,j). Similarly, no increase in IL-1β processing was observed in unprimed *Card9*[−/−] compared with WT BMDCs infected with *S.* Typhimurium (m.o.i. 10) (Supplementary Fig. 6). Taken together, these data support a role for CARD9 in regulating SYK activation of NLRP3 to control caspase-8 processing of IL-1β in the canonical inflammasome in BMDMs.

**SYK controls caspase-8 recruitment to the inflammasome.** How does SYK regulate caspase-8 activity in response to NLRP3 activation? We hypothesized that SYK alters caspase-8 recruitment to the inflammasome speck. To test this hypothesis we stimulated LPS-primed WT, *Card9*[−/−] and *Pycard*[−/−] BMDMs with nigericin in the presence or absence of R406, and stained the cells for active caspase-8 and caspase-1. After fixation and counterstaining with 4,6-diamidino-2-phenylindole, we observed speck-like structures for both caspase-1 and caspase-8 in WT and CARD9 knockout macrophages, but not in *Pycard*[−/−] BMDM (Fig. 7a). Quantification of the number of cells containing specks (1,000 cells per treatment, selected from random fields) confirmed that *Card9*[−/−] BMDMs had more caspase-8-positive specks than WT cells. The number of WT and *Card9*[−/−] BMDM containing caspase-1-positive specks was very similar to each other (Fig. 7c). In the presence of the SYK inhibitor R406 the number of cells containing caspase-8-positive specks was reduced compared with cells without inhibitor, but was the same for WT and *Card9*[−/−] BMDM (Fig. 7b). These observations corroborate the caspase-8 cleavage data (Fig. 2g) and caspase-8 inhibition assays (Fig. 6a–h) supporting the concept that SYK and CARD9 regulate the recruitment of caspase-8 to the inflammasome after NLRP3 activation to process IL-1β.

**CARD9 is a central signalling hub for inflammatory signalling.** Our data identify a novel, inhibitory, role for CARD9 in the regulation of NLRP3-induced IL-1β production. CARD9 regulates inflammatory responses downstream of numerous PRRs including NOD2, Dectin-1 and RIG-I. This suggests that CARD9 may function as a signalling hub coordinating the cellular

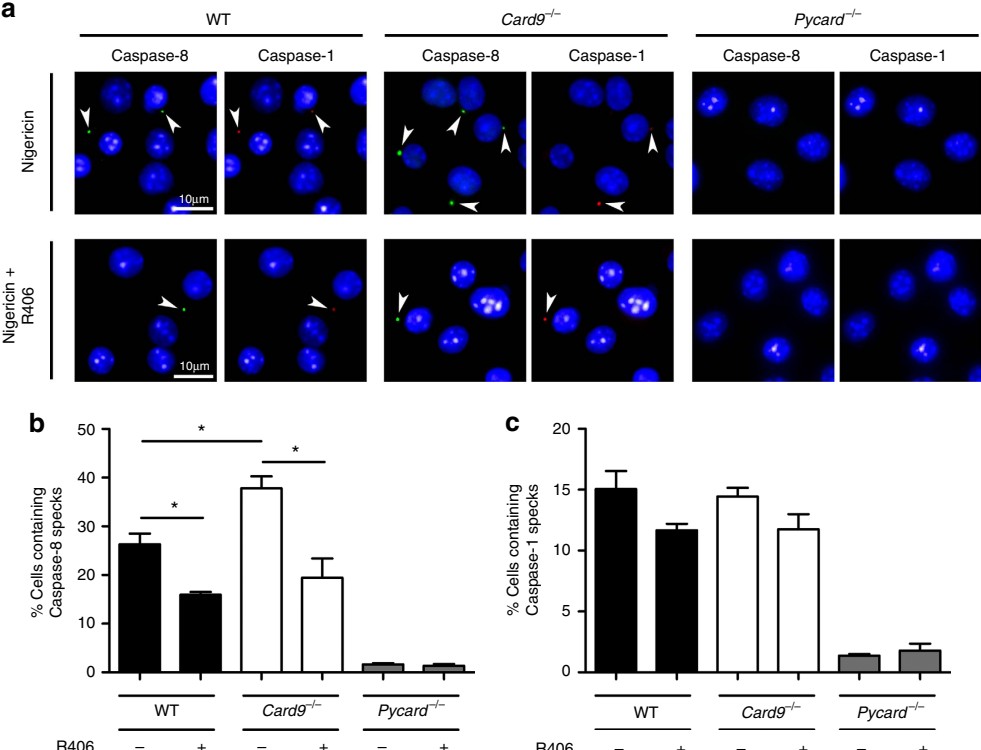

**Figure 7 | SYK activity increases the number of caspase-8 specks in response to NLRP3 stimulation in *Card9*[−/−] BMDMs.** (**a–c**) WT, *Card9*[−/−] or *Pycard*[−/−] LPS-primed BMDMs were incubated with nigericin (5 μM, 30 min) in the presence of caspase-1 and caspase-8 FLICA substrates, with or without the SYK inhibitor R406. (**a**) Pattern of caspase-1 and caspase-8 specks observed by immunofluorescence labelling of the protein. Percentage of cells containing (**b**) caspase-8 or (**c**) caspase-1 specks. *$P < 0.05$ (one-way analysis of variance with Tukey's multiple comparisons test). (**a**) Image is representative of three independent experiments. (**b,c**) Data from three independent experiments (mean and s.e.m.). Scale bar, 10 μm.

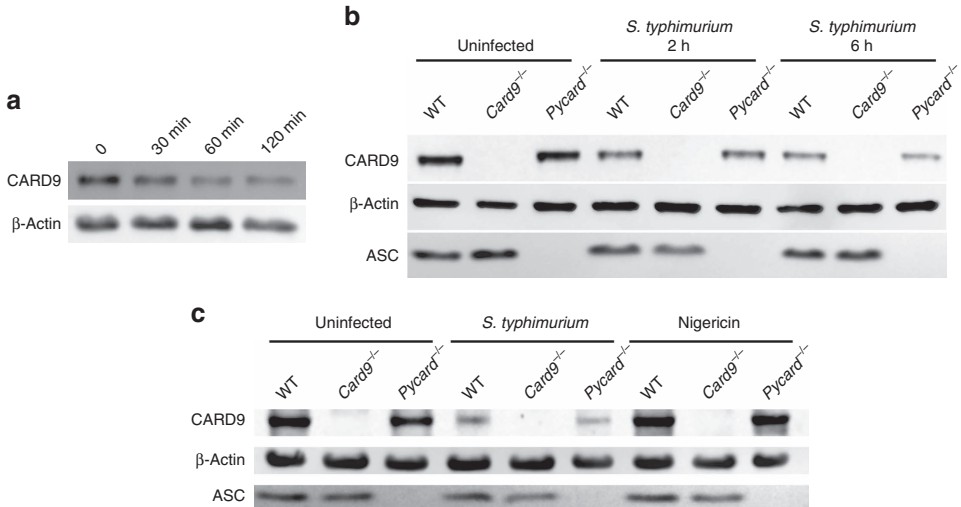

**Figure 8 | CARD9 acts as a regulatory switch during infection.** (**a**) Immunoblotting of CARD9 and ASC in cell lysates from unprimed WT BMDMs after infection with *S*. Typhimurium (m.o.i. 5) for 0, 30, 60 and 120 min. (**b**) Immunoblotting of CARD9 and ASC in cell lysates from unprimed BMDMs (WT, *Card9*$^{-/-}$ or *Pycard*$^{-/-}$) after infection with *S*. Typhimurium (m.o.i. 5) for 0, 2 and 6 h. (**c**) Immunoblotting of CARD9 and ASC in cell lysates from uninfected, *S*. Typhimurium-infected (m.o.i. 5 for 2 h) or nigericin-stimulated (5 μM, 30 min) LPS-primed BMDMs (WT, *Card9*$^{-/-}$ or *Pycard*$^{-/-}$). (**a**–**c**) Images are representative of three independent experiments.

response to infection and permitting a rapid integrated inflammatory response to a range of PRRs. To test this hypothesis we investigated whether *Salmonella* infection alters CARD9 protein expression and thereby potentially alters the resulting level of inflammasome activation. CARD9 is constitutively expressed in many cell types including macrophages[11] and after infection of WT BMDM with *S*. Typhimurium (m.o.i. 10) a progressive decrease in the expression of this protein was observed within 30 min after infection (Fig. 8a). The decrease in CARD9 expression was independent of inflammasome assembly, as similar CARD9 expression profiles were seen in infected LPS-primed or unprimed *Pycard*$^{-/-}$ BMDMs (Fig. 8b). Nigericin stimulation does not trigger CARD9 downregulation (Fig. 8c) supporting our hypothesis that the regulation of CARD9 expression in macrophages is specifically downregulated on *Salmonella* infection possibly to maximize the host inflammasome response to infection.

Our data and other published work supports a role for CARD9 as an important regulator of PRR signalling where it can up- or downregulate pro-inflammatory cytokine production in a pathogen-dependent manner[6,8,32]. If CARD9 is a central regulator of the inflammatory response to pathogens then signalling through other CARD9-dependent PRRs that recognize salmonellae should also be altered in response to infection. NOD2 recruits CARD9 to activate p38 and JNK MAPK signalling[11,33], but NF-κB signalling is activated independently of this adaptor protein. Pro-IL-1β expression is transcriptionally regulated by NF-κB, p38 and JNK MAPK, whereas TNF-α transcription is independent of p38 and JNK activity[34]. We investigated whether CARD9 was important in regulating NOD2-induced signalling in response to *S*. Typhimurium infection. WT and *Nod2*$^{-/-}$ BMDMs were infected with *S*. Typhimurium (m.o.i. of 10), and the levels of IL-1β and TNF-α were determined by ELISA. Infection of *Nod2*$^{-/-}$ macrophages induced a small increase in IL-1β production, but suppressed TNF-α production (Supplementary Fig. 7a–c). These data are consistent with the NOD2-CARD9 axis negatively regulating pro-IL-1β production through an action on MAPK signalling independently of NOD2 induction of TNF-α. Consistent with this idea, overstimulation of NOD2 at

the same time as infection with *S*. Typhimurium leads to a CARD9- and NOD2-dependent decrease in pro-IL-1β expression (Supplementary Fig. 7d). Functional association network analysis showed that the primary interactome of CARD9 consists of key immune and death signalling pathways in both mice and humans (Supplementary Fig. 8). Collectively these data support the idea that CARD9 functions as a signalling hub to co-ordinate inflammatory responses to pathogens such as *Salmonella*.

## Discussion

In this study we have identified a new role for CARD9 as a negative regulator of IL-1β production in response to bacterial infection. This is in stark contrast to the role of CARD9 in fungal signalling where this adaptor is important for driving pro-inflammatory signalling responses to infection[3,6]. The recognition of infection by host cells triggers a series of complex, pathogen-specific, signalling events to drive an appropriate inflammatory response. It is increasingly clear that inflammatory signalling pathways do not function in isolation, but form complex networks in which key constituents act as nodes or hubs linking apparently distinct pathways. The data we present here, along with that of other published studies, clearly indicate that CARD9 is one such protein. CARD9 coordinates Toll-like receptor-dependent and independent NF-κB signalling[6,9,11,35], reactive oxygen species production[10], autophagy[36], non-canonical inflammasome function[3] and, as we now demonstrate, canonical inflammasome assembly in a context-dependent manner. Importantly, the specific role of CARD9 differs between cell types, thereby contributing a further level of regulatory control in enabling cell specificity in the host response to infection [32,35,37].

Our data show CARD9 fine-tunes IL-1β production in two distinct ways (Supplementary Fig. 9). During 'signal 1', CARD9 specifically downregulates pro-IL-1β transcription without affecting other NF-κB target genes. This occurs following activation of p38 and JNK signalling via NOD2, and is consistent with reports that NOD2 can have specific inhibitory roles that are dependent on the context of its activation[38–43]. How CARD9 regulates pro-IL-1β in this context remains to be fully elucidated,

but is likely to involve the transcription factor AP-1. AP-1 is a known transcriptional regulator of pro-IL-1β (ref. 44), which is itself regulated by p38 and JNK activity[45]. There is, therefore, increased expression of pro-IL-1β during the 'signal 1' phase of inflammasome stimulation in Card9[−/−] cells. However, in WT and Card9[−/−] LPS-primed macrophages, which express similar levels of pro-IL-1β, the production of IL-1β is further increased in Card9[−/−] compared with WT cells. This suggests the elevated production of IL-1β from Card9[−/−] cells is not solely due to the increased pro-IL-1β substrate availability, but is also due to an effect of CARD9 during 'signal 2' of inflammasome activation. During 'signal 2', CARD9 suppresses SYK phosphorylation, which consequently reduces ASC phosphorylation and the subsequent assembly and activation of the inflammasome. In macrophages, p-SYK phosphorylates ASC, enhancing NLRP3 inflammasome activity[24,25], whereas in dendritic cells SYK has no effect on the canonical NLRP3 inflammasome[24]. Consistent with these studies our data clearly show that CARD9 negatively regulates the activity of NLRP3 in macrophages, but not in dendritic cells. Our data suggest that, in addition to the well-described regulatory role for CARD9 downstream of SYK, it is also possible that CARD9 can act upstream of SYK during bacterial infection. The regulation of SYK phosphorylation by CARD9 specifically alters caspase-8 recruitment to the inflammasome explaining why we see changes in IL-1β production independently of any effect on pyroptosis. In Salmonella infection caspase-8 processes IL-1β but does not affect pyroptosis[17]. Precisely how ASC phosphorylation enhances caspase-8 recruitment is unclear, but as SYK phosphorylates ASC on its CARD domain[24] it is plausible that altering the pattern of ASC phosphorylation could impair the CARD/CARD interaction between ASC and caspase-1 (ref. 46) and favour interactions between the ASC PYD and Caspase-8 DED[47].

Identification of negative regulators of inflammasome activation may have important clinical implications because dysregulated inflammasome activity is associated with a number of important diseases[2,48]. Genome-wide association studies found strong correlations between loss-of-function CARD9 mutations and an increased likelihood of developing inflammatory diseases[12–16]. In a M. tuberculosis infection model CARD9 knockout mice have an increased bacterial burden and develop exacerbated systemic inflammatory responses[27], further strengthening the link between CARD9 and inflammatory diseases. Similarly, Card9[−/−] mice are deficient in controlling Candida albicans infection, a fungal pathogen capable of stimulating NLRP3 (ref. 6). These in vivo and clinical observations emphasize CARD9 role as a negative regulator for inflammation, possibly by fine-tuning NLRP3-mediated IL-1β production. In conclusion we have identified a novel negative regulatory role for CARD9 on IL-1β production in macrophages by modulating pro-IL-1β expression and caspase-8 recruitment to the inflammasome in response to bacterial infection. These data support a role for CARD9 as a central signalling hub in co-ordinating innate immune inflammatory responses to infection.

## Methods

**Mice.** WT C57BL/6 mice were obtained from Charles River, UK. Nlrc4[−/−], Nlrp3[−/−] and Pycard[−/−] mice on a C57BL/6 background were produced by Millenium Pharmaceuticals and obtained from Kate Fitzgerald (University of Masschusetts). Card9[−/−] mice on a C57BL/6 background, originally produced by Xin Lin (University of Texas)[11], were provided by David Underhill (Cedars-Sinai Medical Center). Nod2[−/−] mice were provided by Peter Murray (St. Jude Children's Research Hospital). Mice were backcrossed on a C57BL/6 background at least eight generations. All mice strains were bred independently. All work involving live animals complied with the University of Cambridge Ethics Committee regulations and was performed under the Home Office Project License number 80/2572.

**In vivo infections.** S. Typhimurium M525P was grown statically for 18 h at 37 °C in LB Broth (Sigma), then washed and resuspended in PBS (Sigma). 1.90–2.04 × 10^4 colony-forming units per mouse were administered systemically to 8- to 16-week-old mice, both male and female, via the lateral tail vein, while control mice were inoculated with PBS only. Mice were killed at days 1, 3 and 7 after infection and their spleens and livers were aseptically removed. Organs were homogenized in 10 ml sterile water using a Colworth stomacher. Organ homogenates were then 10-fold serially diluted in PBS, plated on LB agar plates and incubated overnight at 37 °C followed by enumeration of colony-forming units.

To obtain splenocyte cell suspensions, spleens were disrupted through a 70 µm cell strainer (BD Biosciences), washed in RPMI containing 2% Hyclone, resuspended in Red Blood Cell lysis buffer (Sigma) and incubated for 10 min at room temperature. The suspension was then centrifuged at 300g for 10 min and the pellet washed twice in RPMI. The purified splenocytes were lysed and the proteins probed by immunoblotting as described below.

All mice were housed in a specific pathogen-free facility and all work involving live animals complied with the University of Cambridge Ethics Committee regulations under Home Office Project License number 80/2572.

**Cell culture and stimulation.** Primary BMDMs were prepared and cultured for 6–9 days as previously described[17]. BMDCs were prepared and cultured for 7–10 days as previously described[49]. Briefly, mice were killed by cervical dislocation, the skin was sterilized with ethanol (70%) before removal of the leg. The tibia and femur were removed, cleaned of muscle and the proximal and distal epiphysis cut away. For BMDMs, the bone marrow was flushed out using DMEM supplemented with 10% FCS (Thermo Fisher Scientific), 20% L929 conditioned media and 5 mM L-glutamine (Sigma). For BMDCs, the bone marrow was flushed out using BMDC growth media (RPMI 1640 supplemented with 10% FCS, 50 µM 2-mercaptoethanol and 1,000 U ml^−1 granulocyte–macrophage colony-stimulating factor (Thermo Fisher Scientific). The bone marrow cells were centrifuged (300g for 10 min at 15 °C) and resuspended in appropriate growth media. Cells were cultured at 37 °C in 5% CO_2 and growth media replaced every 2 days. Infection of BMDM with S. Typhimurium strain SL1344 and ΔfliCΔfljBΔprgJ was performed as previously described[17]. For NLRP3 inhibition assay, the cells were pre-incubated with glibenclamide (Sigma) 200 µM, MCC950 10 µM (Cayman Chemical) or vehicle control (dimethylsulfoxide) for 15 min and infection was carried out as described above. For NOD2 co-stimulation assay, the cells were infected with S. Typhimurium in the presence or absence of muramyl dipeptide 10 µg ml^−1 (Invivogen).

Selected experiments required BMDMs primed with LPS. This was performed by incubating the cells in growth media containing 200 ng ml^−1 ultrapure LPS from E. coli O111:B4 (InvivoGen) for 3 h at 37 °C and 5% CO_2, followed by washes in media alone. For stimulation experiments, the LPS-primed BMDMs were incubated with nigericin 10 µM (Sigma), 5 mM ATP (Sigma) or 60 ng ultrapure flagellin from S. Typhimurium (Invivogen), after incubation at room temperature for 20 min with Profect-P1 reagent (Target Systems) for transfection complex formation. AIM2 stimulation was performed using Poly(dA:dT)/LyoVec (Invivogen) 2 µg ml^−1 for 4 h. Caspase-8 inhibition experiments were performed in LPS-primed cells by the addition of the stimuli with or without 10 µM Z-IETD-FMK (MBL International). SYK inhibition experiments were conducted by pre-incubating the LPS-primed cells for 1 h with or without 1 µM R406 (Invivogen), followed by stimulation in presence or absence of 1 µM R406.

**Quantitative PCR.** Following stimulation, cells were treated with RNAprotect Cell Reagent (Qiagen) and total RNA was isolated using the RNeasy Mini kit (Qiagen) according to the manufacturer's instructions. Genomic DNA was removed using the TURBO DNA-free kit (ThemoFisher Scientific). The primers used (Supplementary Table 1) were selected based on data submitted to the primer bank database (http://pga.mgh.harvard.edu/primerbank/index.html)[50]. The quantitative RT–PCR was performed with the SensiFAST One-Step Real-Time PCR kit (Bioline) using a Rotor-Gene Q real-time PCR cycler (Qiagen). Data analysis was carried out using the mean of glyceraldehyde 3-phosphate dehydrogenase and β-actin as reference genes, using Pfaffl method to correct for reaction efficiency[51].

**Immunoprecipitation, protein precipitation and immunoblot.** Cell culture supernatant and spleen homogenate proteins were precipitated using methanol and chloroform. Briefly, a volume of the sample was vortexed with one volume of ice-cold methanol and 0.25 volume of chloroform, followed by centrifugation (4 °C, 16,000g, 12 min). The intermediate phase was collected, washed two times in ice-cold methanol and resuspended in Pierce Lane Marker Reducing Sample Buffer (Life Technologies).

Cells were lysed in lysis buffer containing 10 mM Tris (pH 7.4), 150 mM NaCl, 5 mM EDTA, 1% Triton X-100, 10 mM NaF, 1 mM NaVO_4, 20 mM phenylmethylsulfonyl fluoride, phosphatase inhibitor cocktail 3 (1 in 100 dilution, Sigma) and protease inhibitor cocktail (1 in 100 dilution, P8340, Sigma). Protein levels were quantified using Pierce BCA Protein Assay Kit (Life Technologies). The samples were used for co-immunoprecipitation or incubated for 5 min at 100 °C with Pierce Lane Marker Reducing Sample Buffer (Life Technologies) for immunoblotting.

For co-immunoprecipitation analysis cells were reversibly crosslinked with 5 mM DTBP (Fisher Scientific) for 30 min at 4 °C then lysed as described above. Protein concentration was adjusted to 600 µg ml$^{-1}$, and 800 µl was incubated overnight with 2 µl antibody (goat anti-ASC (sc-33958, Santa Cruz), rabbit anti-CARD9 mouse (12283, Cell Signaling) and mouse anti-SYK (MA1-19332, Thermo Scientific)) and 20 µl of Protein A/G PLUS-agarose beads (Santa Cruz). The remainder of the protein sample was stored for immunoblotting. After incubation, the beads were washed four times in lysis buffer, ressuspended with Pierce Lane Marker Reducing Sample Buffer (Life Technologies), incubated for 5 min at 100 °C, centrifuged for 1 min at 5,000$g$ and the supernatant used for immunoblotting.

Immunoblots were probed using the following primary antibodies: caspase-1 p10 (mouse) (sc-514, Santa Cruz) 1 in 500; cleaved caspase-8 (rabbit) (8592, Cell Signaling) 1 in 500; caspase-8 (rabbit) (IG12, Enzo) 1 in 500; IL-1β (goat) (AF-401, R&D Systems) 1 in 1,000; SYK (mouse) (MA1-19332, Thermo Scientific) 1 in 1,000; p-SYK (rabbit) (2711, Cell Signaling) 1 in 1,000; ASC (rabbit) (AL177, Enzo) 1 in 2,000; B-Actin (mouse) (AB3280, ABCAM) 1 in 2,500; and CARD9, mouse preferred (rabbit) 1 in 1,000 (12283, Cell Signaling). The secondary antibodies used were as follows: anti-goat IgG-HRP (sc-2922, Santa Cruz) 1 in 5,000; anti-mouse IgG-HRP (7076, Cell Signaling) 1 in 6,000; and anti-rabbit IgG-HRP (A24537, Thermo Scientific) 1 in 6,000 as appropriate. Uncropped scans of immunoblots are provided as Supplementary Figs 10–22.

**Immunofluorescence microscopy.** To label CARD9 and ASC, stimulated cells were fixed in − 20 °C methanol for 5 min, after washing with PBS, non-specific labelling was blocked by incubation at 37 °C for 1 h in 10% normal goat serum (Dako) containing 0.1% saponin (Sigma). ASC was labelled with anti-ASC primary (1:500 dilution, AL177; Enzo) and Alexa-fluor 568 secondary antibody (1:1,000 dilution, anti-rabbit Alexa-fluor 568 (Invitrogen)). To label CARD9 the cells were subsequently blocked in PBS containing 1% BSA and 0.1% saponin. Immuno-labelling was performed with anti-CARD9 (1:300 dilution, sc-49408, Santa Cruz) and anti-Rabbit-Alexa 430 (1:1,000 dilution, Invitrogen). Cells were counterstained using 4,6-diamidino-2-phenylindole mounting medium (Vecta Labs).

Speck enumeration was performed in cells stimulated with nigericin in presence of FAM-FLICA Caspase 8 Assay Kit, green (1:150, Immunochemistry Technologies) and FAM-FLICA Caspase 8 Assay Kit, red (1:150, Immunochemistry Technologies), followed by fixation with 4% paraformaldehyde for 15 min at room temperature. Non-specific labelling was prevented by incubating cells with 10% normal goat serum (Dako) and 0.1% saponin (Sigma) for 1 h. In all, 1,000 cells per treatment were selected from random fields and the specks counted.

**Cellular viability assays and intracellular bacteria counts.** Cytotoxicity was quantified by measuring LDH activity after being released from live cells. Uninfected BMDMs (ranging from 12,500 to 200,000 cells per well) were used as standards. After treatment, the cells were washed three times with pre-warmed PBS and the intracellular LDH was released by lysing the cells with Triton X-100 1.2% for 1 h at 37 °C. LDH activity was then measured using CytoTox 96 Non-Radioactive Cytotoxicity Assay (Promega)[17]. Cellular viability was them calculated in relation to the uninfected control containing 200,000 cells (100% viability). Intracellular bacteria counts were performed using LB agar plates after overnight incubation.

**Cytokine quantification.** Secreted cytokines were measured in the culture supernatants. All cytokines were measured according to the manufacturer's instructions. For IL-1β OptEIA Mouse IL-1β Set (BD Biosciences) was used and TNF-α using the DuoSet ELISA kit (R&D Systems).

**Computational analysis.** Functional association network analysis was performed using STRING v10 (ref. 52) and either murine or human CARD9 as search proteins. Primary interactomes were initially established using a high-confidence cutoff score of 0.7 and a maximum of 50 interaction partners. These interactomes were further expanded by relaxing the inclusion criteria to a cutoff value of 0.4. Data are presented with the thickness of the line between protein nodes representing the confidence level associated with that interaction.

**Statistical analysis.** Statistical significance was determined by one-way analysis of variance with Tukey post-comparison tests using a confidence interval of 95%. In vivo data were analysed by two-way analysis of variance with Bonferroni post-test using a confidence interval of 95%.

**Data availability.** The data that support the findings in this study are available in the University of Cambridge data repository and are available with the doi 10.17863/CAM.778 (http://dx.doi.org/10.17863/CAM.778).

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

## Acknowledgements

We thank Dr. Martyn F. Symmons for his suggestions and Caroline R. Paié for the drawing used in the supplementary diagram. M.P. was supported by CAPES (Coordenação de Aperfeiçoamento de Pessoal de Nível Superior, Brazil). This work was supported by a grant from the BBSRC BB/K006436/1 and a Wellcome Trust Investigator 108045/Z/15/Z award to C.E.B.

## Author contributions

The experiments were performed and the results analysed by M.P. and P.T. The experiments designed by M.P., P.T., J.A.W. and C.E.B. Manuscript written by M.P., J.A.W., T.P.M. and C.E.B.

## Additional information

**Competing financial interests:** The authors declare no competing financial interests.

**How to cite this article**: Pereira, M. *et al.* CARD9 negatively regulates NLRP3-induced IL-1β production on Salmonella infection of macrophages. *Nat. Commun.* 7:12874 doi: 10.1038/ncomms12874 (2016).

