## [Peer review file · Nature Communications]

Reviewers' comments:

Reviewer #1 (Remarks to the Author):

This paper sets out to examine the role of CARD9 in regulating inflammasome signaling during infection with *Salmonella typhimurium*. The authors show that CARD9 down-regulates IL1 β production following *Salmonella* infection through reduction of pro-IL1 β induction, inhibiting SYK-dependent NLRP3 activation and reducing caspase-8 recruitment to the inflammasome. This is a novel pathway involved in the regulation of inflammasome assembly during infection. The findings are important and of general interest.

Major Points

1. The changes in pro-caspase8 and active caspase-8 shown in Fig 2g are relatively small and in particular the band shown for the caspase-8 p18 subunit is rather faint. Densitometric scanning data from the repeated experiments would strengthen their conclusions.
2. In figure 5a, there is some CARD9 staining in the untreated Card9 $^{-/-}$ cells - presumably background but needs some elaboration/explaining to be certain the other CARD9 staining is specific. Additionally, there are clear CARD9 specks in the Pycard $^{-/-}$ cells but I can't see these in the WT cells and thus could not comment on whether they co-localise with ASC specks.
3. In figure 5b, the differential recovery of SYK and phospho-SYK from the CARD9 and ASC immunoprecipitates is important to the main argument of the paper. The quality of these blots makes the data difficult to interpret and clearer immunoblots would allow better evaluation of the assertion that CARD9 preferentially associates with non-phosphorylated SYK.
4. The effect of Z-IETD-FMK on IL1 β secretion from infected Card9 $^{-/-}$ macrophages suggests Caspase-8 is important in the CARD9 mediated effects. However, knock-down of Caspase8 would be more specific than a drug inhibitor and strengthen the conclusions.
5. The effect of CARD9 in *Salmonella* infection is clear, but I think the title may be a little misleading to extend these findings to bacterial infection in general.

Minor Points

1. The legends for Supplementary Figures 5 and 6 seem to be reversed.
2. The ordering of the panels in many of the figures is not logical and differs from figure to figure; it would be easier to follow if the same scheme was adopted for each figure. I also think the time course data would be better visualized as line graphs which would cut down the number of panels.

Reviewer #2 (Remarks to the Author):

Card9 is critically involved in the signalling of CLRs and several intracellular pattern recognition receptors. Card9-dependent pro-inflammatory cytokine production is protective during fungal infection because it promotes effective pathogen elimination. Though Card9 seems to be necessary for the production of certain pro-inflammatory cytokines in several biological contexts, a less-explored avenue of investigation is that Card9 may negatively regulate inflammatory signaling in other contexts.

Pereira et al. demonstrate in this study that Card9 negatively regulates proIL-1 β expression and Nlrp3 inflammasome activation upon Gram negative bacterial infection of macrophages, resulting in up to 3-fold more IL-1 β production from Card9-deficient cells. The precise mechanism by which

Card9 negatively regulates proIL-1b expression still needs to be investigated in detail, but it may in part be at the level of NOD2/JNK/p38 signalling. Part of the increase in IL-1b secretion in Card9-deficient cells is likely a reflection of increased proIL-1b expression. However, the authors provide evidence suggesting that enhanced Syk and Caspase-8 activity in Card9-deficient cells also contributes to enhancement of IL-1b processing and secretion. Overall, they identify a novel function of Card9 as a negative regulator of IL-1b upon bacterial infection of macrophages. The experiments appear to have been performed diligently, with care taken to ensure their reproducibility. I suggest a few experiments below that might improve the study. I have some disagreements on interpretation of the data, but most of those can be resolved by adjusting the wording.

Major

1) Role of Caspase-1 and Caspase-8

The authors state 1) "The mature form of caspase-8 was increased in Card9^{-/-} BMDMs at 2 hours (Fig. 2g) suggesting that, in the absence of CARD9, increased caspase-8 activity is responsible for the increased conversion pro-IL-1 β to its mature form." And also 2) "The enhanced production of NLRP3-dependent IL-1 β secretion in Card9^{-/-} BMDMs is independent of caspase-1 (Fig. 2g)". I disagree with this interpretation of the data. Even if the cleavage of Casp1 and Casp8 remained the same in Salm-stimulated Card9 KO (vs wt), an increase in proIL-1b expression should itself lead to an increase in mature IL-1b (basic enzyme kinetics - more substrate, more product). In my opinion, the authors should instead write 1) that an increase in Casp8 cleavage may contribute to enhanced IL-1b processing, and 2) that the enhanced production of NLRP3-dependent IL-1 β secretion in Card9^{-/-} BMDMs may involve Casp8. If they want to state that the increase in IL-1b secretion in Card9 KOs is truly independent of Casp1, then they should provide experimental evidence for this (e.g. using a Casp1 inhibitor - YVAD or VX-765 - in experiments such as those in Figure 6).

2) Is it really deficiency of Card9 that accounts for the increase in Salm-induced proIL-1b observed in BMDMs? Were the wt and Card9-deficient mice used for these experiments derived from homozygous or heterozygous breedings? There have been several papers recently demonstrating that certain passenger mutations acquired during inbreeding can have a frightfully large effect on in vivo and in vitro phenotypes (e.g. mutations in various DOCK genes in inbred ASC KO and NLRP10 KO lines). If het x het breedings were used and the mutation is not genetically linked (i.e. not on the same region of the chromosome as Card9) then WT and KO littermates would have an equal chance of inheriting any genetic difference acquired during inbreeding. If WT and Card9 KO mice were bred independently (especially for many generations), then there is a risk that differences in the genetic drift/passenger mutations between the WT and KO lines (rather than Card9-deficiency) may account for the phenotype the authors observe. If the authors have used het x het breedings to obtain wt controls, then I am reasonably convinced that the phenotype observed is truly because of Card9 deficiency. If not, the authors have to either confirm their results using littermate controls, or reconstitute Card9 in Card9-deficient cells (e.g. with retroviruses) to show that this normalizes proIL-1b levels and IL-1b secretion. Regardless of how WT controls were produced, this information has to be provided in the Materials and Methods.

3) Is this a cell-intrinsic phenotype? In other words, does Card9 directly inhibit pathways controlling IL-1b expression and directly inhibit Syk phosphorylation? Supplementary Figure 7 suggests the authors favour this interpretation, presumably on the basis of their immunoprecipitation data which I did not find especially compelling. A possible contribution of autocrine cytokine signalling or other indirect actions of Card9 should not be overlooked. Card9-deficient neutrophils fail to secrete IL-10 in response to Mycobacteria (Dorhoi J Exp Med 2010). The same may be true for macrophages, and lack of autocrine IL-10 signalling could in principle influence pro-IL-1b expression and Nlrp3 activation.

4) The defect in proIL-1b expression is an interesting finding but it requires mechanistic

explanation. Card9 is known to promote proIL-1b production in response to Syk-dependent (e.g. CLR ligation) and independent (e.g. Rad50, RIG-I) pathways. What is so different about pathways leading to proIL-1b expression in response to Gram negative bacteria that Card9 plays an opposite role?

5) I do not find data in figures 8, S5, or S6 to be compelling evidence that "CARD9 is a central signaling hub for inflammatory signaling". The data is consistent with this proposition, but the proposition itself and the abundant published data to support it are not new. The novelty of this study is that it proposes that Card9 can also negatively regulate pro-inflammatory cytokine production. On a similar note I find the title long, uninformative, and not reflective of the central finding of the paper. These decisions I leave to the authors and editors, but a simple alternative would be 'Card9 negatively regulates Nlrp3-induced IL-1b production upon bacterial infection of macrophages'.

Minor

6) In the introduction, the authors state while referring to previous publications that "CARD9 regulates SYK activity and this kinase phosphorylates the CARD domain of ASC when NLRP3, but not NLRC4, is activated to increase IL-1 β and IL-18 production[22-25]". In the CLR signalling pathway, it is thought that Card9 acts downstream of Syk - Syk is phosphorylated at CLRs, Syk phosphorylates PKCs, and PKCs phosphorylate and activate Card9 to trigger recruitment of Bcl10/Malt1 and activation of NF-kB. As far as I am aware, the authors are the first to provide evidence that Card9 can also act upstream of Syk (directly or indirectly), but what they have written above makes it seem like regulation of Syk by Card9 was already known.

7) The in vitro bacterial counts (Figure S1a-e) should be presented on a linear scale for each MOI.

8) The LDH assay is an assay for lytic cell death, but the authors present the data as viability. It should be briefly indicated in the Materials and Methods how this was calculated, and what was used to determine 0% and 100% viability. How was the potential contribution of bacterial LDH accounted for?

9) Glibenclamide can inhibit Nlrp3 activation, but must be used at exceedingly high concentrations that in our hands also inhibit secretion of TNF. I can accept that IL-1b secretion in WT cells is mostly Nlrp3-dependent, but the authors have shown (Figure 3f, and Man et al 2014) that Nlrp3 also contributes to IL-1b secretion in response to WT Salm. Therefore, if glibenclamide is truly inhibiting Nlrp3, I find it is also somewhat unexpected it has no effect on IL-1b secretion by WT cells (only by Card9 KO). I would suggest that the authors reproduce these results with MCC950/CRID3, since it is a more potent and specific inhibitor of Nlrp3 than glibenclamide.

10) Did the authors perform isotype control IPs to determine the (co-)immunoprecipitation is specific? Except for the interaction of ASC with p-Syk, The IP data in Figure 5 is not especially convincing. The pSyk data in Card9 KOs is also not especially compelling.

11) The authors state "BMDM infection with *S. typhimurium* (MOI 10) in the presence of the caspase-8 inhibitor Z-IETD-FMK, as expected, had no effect on cell viability (Fig. 6a) because caspase-8 does not induce pyroptosis in response to infection with this pathogen." They observed similar results are observed with Nigericin. To say that the caspase-8 inhibitor is not influencing viability (using LDH release after lytic death as a read-out) may not be entirely correct, because it may reduce apoptosis without influencing lytic/pyroptotic death (Sagulenko et al CDD 2013).

12) As the authors point out, several GWAS studies have associated hypomorphic alleles of CARD9 with pro-inflammatory diseases, suggesting that CARD9 may negatively regulate inflammation. There is also evidence in mice that Card9 can negatively regulate inflammatory responses. For

instance, Card9-deficient mice fail to control Mycobacterium infection, and display an exacerbated (and lethal) inflammatory phenotype (Dorhoi J Ex Med 2010). The failure to effectively control the initial Mycobacterial infection may result from lack of Card9-dependent pro-inflammatory pathways. However, the late hyperinflammatory response may arise from failure of Card9-deficient neutrophils to produce the anti-inflammatory IL-10 (meaning that Card9-deficient neutrophils have a hyperinflammatory phenotype). Similar results are seen upon Listeria infection of Card9-deficient mice (Hsu Nat Immunol 2010). Could the authors' finding that Card9 negatively regulates pro-inflammatory cytokine production be relevant in the context of these previous publication?

Reviewer #3 (Remarks to the Author):

Pereira and colleagues demonstrate a novel inhibitory role for CARD9 in IL-1beta production. The authors show that CARD9 reduces SYK mediated NLRP3 inflammasome activation, which in turn leads to a reduction in caspase-8 activity.

The study is well performed and the results appropriately interpreted. This regulatory role for CARD9 in IL-1beta production is both interesting and novel. I do however have a few concerns/comments:

1. Is there biological relevance of this CARD9 mediated regulation of IL-1beta in the setting of salmonella infection in vivo? For salmonella infection the authors show that there is no clear difference in bacterial burdens between WT and Card9^{-/-} (Supp. Fig 1f,g). In addition the increase in pro-IL-1beta appears rather modest (Fig 1p); densitometry of this blot would be helpful.
2. The title is somewhat misleading as other NLRP3 agonists seem to mediate a similar response (ie. Nigericin) and hence this is not restricted to bacterial infections. Have the authors examined other NLRP3 agonists to confirm this can be extrapolated to both soluble and crystalline NLRP3 agonists? In addition have the authors examined if the AIM2 inflammasome is affected?
3. Can the authors show biological relevance for this pathway utilizing another NLRP3 agonist in vivo?

We would like to thank the referees for their reviews of our manuscript and providing us with their constructive comments on our work. We have made extensive revisions to our MS to address as fully as possible all of their comments and we think our paper is now much stronger as a result.

Reviewers' comments:

Reviewer #1 (Remarks to the Author):

This paper sets out to examine the role of CARD9 in regulating inflammasome signaling during infection with Salmonella typhimurium. The authors show that CARD9 down-regulates IL1beta production following Salmonella infection through reduction of pro-IL1beta induction, inhibiting SYK-dependent NLRP3 activation and reducing caspase-8 recruitment to the inflammasome. This is a novel pathway involved in the regulation of inflammasome assembly during infection. The findings are important and of general interest.

Major Points

1. The changes in pro-caspase8 and active caspase-8 shown in Fig 2g are relatively small and in particular the band shown for the caspase-8 p18 subunit is rather faint. Densitometric scanning data from the repeated experiments would strengthen their conclusions.

We have performed densitometric analysis of all of our blots and we have included the data in figure 2 of our revised MS. For clarity the western blot with the added densitometric scanning data is reproduced in figure 1 below.

Figure 1: CARD9 modulates pro-IL-1β expression. (a) Expression of pro-IL-1β, SYK, pro-caspase-8, pro-caspase-1, CARD9, ASC, β-actin in cell lysates and caspase-1 p10, caspase-8 p18 and IL-1β in culture supernatants from WT, Card9^{-/-} and Pycard^{-/-} BMDMs after 2 or 6 hours of infection with S.

Typhimurium (MOI 5). Relative densitometry analysis of protein expression (*Card9*^{-/-}/WT) in culture supernatants (b) or cell lysates (*Card9*^{-/-}/WT, actin-normalized) (c).

2. In figure 5a, there is some CARD9 staining in the untreated *Card9*^{-/-} cells - presumably background but needs some elaboration/explaining to be certain the other CARD9 staining is specific. Additionally, there are clear CARD9 specks in the *Pycard*^{-/-} cells but I can't see these in the WT cells and thus could not comment on whether they co-localise with ASC specks.

To answer the second part of the referee's comments first we do see CARD9 specks in the WT cells. We have now included arrows indicating the CARD9 specks as well as the ASC specks in both WT and *Pycard*^{-/-} cells to clarify this issue. We do not see CARD9 specks, but we do see ASC specks, in the *Card9*^{-/-} cells which suggests that the antibody shows specificity for CARD9 although there is clearly some non-specific back ground staining in these cells. The antibody is highly specific in our western blot analysis which also supports the concept that there is selectivity for CARD9 (Figure 2 shown below). To address this point, we have changed the image (shown below as Figure 3, Figure 5a in the revised manuscript) and the text of the MS as follows:

"Immunolocalisation of endogenous CARD9 suggests that this protein forms aggregates in stimulated and unstimulated WT and *Pycard*^{-/-} BMDMs, but it does not co-localize with ASC specks suggesting that CARD9 is not recruited to the ASC speck (**Fig. 5a**). *Card9*^{-/-} BMDMs show some non-specific background staining, but it is much fainter than the CARD9 immunolocalisation in WT and *Pycard*^{-/-} cells."

Figure 2: Anti-CARD9 antibody shows specificity for CARD9 in western blot.

Figure 3: (a) CARD9 and ASC were immuno-labelling in WT, *Card9*^{-/-} and *Pycard*^{-/-} LPS-primed macrophages after nigericin stimulation (5 μM, 30 minutes). Green arrows indicates CARD9 aggregates. Red arrows indicates ASC specks.

3. In figure 5b, the differential recovery of SYK and phospho-SYK from the CARD9 and ASC immunoprecipitates is important to the main argument of the paper. The quality of these blots makes the data difficult to interpret and clearer immunoblots would allow better evaluation of the assertion that CARD9 preferentially associates with non-phosphorylated SYK.

We have repeated the ASC and CARD9 co-Immunoprecipitation assays and obtained better-quality images (Figure 4 below). These images have been used to replace those in the original submission and are included in Figure 5b of the revised manuscript.

Figure 4: Co-immunoprecipitation of ASC and CARD9 from cell lysates of uninfected, *S. Typhimurium* (MOI 10, 30 minutes) infected or nigericin (10 μM, 30 minutes) stimulated LPS-primed BMDMs.

4. The effect of Z-IETD-FMK on IL1beta secretion from infected *Card9*^{-/-} macrophages suggests Caspase-8 is important in the CARD9 mediated effects. However, knock-down of Caspase8 would be more specific than a drug inhibitor and strengthen the conclusions.

We agree that these additional experiments would strengthen the conclusion that caspase-8 is important in the CARD9 mediated effects. We have had little success using siRNA approaches, so we decided to perform additional experiments using CRISPR/Cas9 to knock out caspase 8 from immortalized WT and *Card9*^{-/-} knockout bone-marrow derived macrophages (iBMDM). Prior to commencing this work we first fully characterised the *Card9*^{-/-} iBMDM cells and found they had reduced expression of SYK (see Figure 5 below) which is a key protein in our proposed model so it was not practical to take this work further. We have, however, measured the amount of active caspase-8 by cleavage using western blots, quantified caspase-8 speck number using immunofluorescence analysis and determined the level of caspase 8 gene transcription using qPCR. These assays, along side the caspase 8 inhibitor analysis, all support the conclusion of our study that caspase 8 is playing a role in CARD9 mediated effects. We have amended the MS as follows to clarify this point:

“In the presence of the SYK inhibitor R406 the number of cells containing caspase-8 positive specks was reduced compared to cells without inhibitor, but was the same for WT and *Card9*^{-/-} BMDM (Fig. 7b). These observations corroborate the caspase-8 cleavage data (Fig. 2g) and caspase-8 inhibition assays (Fig. 6a-h) supporting the concept that SYK and CARD9 regulate the recruitment of caspase-8 to the inflammasome after NLRP3 activation to process IL-1β.”

Figure 5: SYK expression in WT and *Card9*^{-/-} iBMDM cells

5. The effect of CARD9 in *Salmonella* infection is clear, but I think the title may be a little misleading to extend these findings to bacterial infection in general.

We have changed the title as suggested to “CARD9 negatively regulates NLRP3-induced IL-1 β production upon *Salmonella* infection of macrophages”

Minor Points

1. The legends for Supplementary Figures 5 and 6 seem to be reversed.

We apologise for this error and have corrected the legends.

2. The ordering of the panels in many of the figures is not logical and differs from figure to figure; it would be easier to follow if the same scheme was adopted for each figure. I also think the time course data would be better visualized as line graphs which would cut down the number of panels.

We have altered the ordering of panels in our figures 1 and 6 to hopefully make them more consistent and easier to follow. We have tried converting the time course data to line graphs, but it makes the data harder to follow in some cases (see figure 6a below for an example of the problem) so we hope the referee will agree with us that the original presentation is clearer (shown below in figure 6b-d).

Figure 6: (a) Cellular viability (as measured by LDH release) of WT, *Nlr4*^{-/-} and *Card9*^{-/-} BMDMs after infection with *S. Typhimurium* SL1344 at MOIs 1, 10 and 50 for 2, 6 and 24 hours. (b-d) Cellular viability (as measured by LDH release) of WT, *Nlr4*^{-/-} and *Card9*^{-/-} BMDMs after infection with *S. Typhimurium* SL1344 at MOIs 1, 10 and 50 for 2 (b), 6 (c) and 24 hours (d).

Reviewer #2 (Remarks to the Author):

Card9 is critically involved in the signalling of CLRs and several intracellular pattern recognition receptors. *Card9*-dependent pro-inflammatory cytokine production is protective during fungal infection because it promotes effective pathogen elimination. Though *Card9* seems to be necessary for the production of certain pro-inflammatory cytokines in several biological contexts, a less-explored avenue of investigation is that *Card9* may negatively regulate inflammatory signaling in other contexts.

Pereira et al. demonstrate in this study that *Card9* negatively regulates proIL-1 β expression and Nlrp3 inflammasome activation upon Gram negative bacterial infection of macrophages, resulting in up to 3-fold more IL-1 β production from *Card9*-deficient cells. The precise mechanism by which *Card9* negatively regulates proIL-1 β expression still needs to be investigated in detail, but it may in part be at the level of NOD2/JNK/p38 signalling. Part of the increase in IL-1 β secretion in *Card9*-deficient cells is likely a reflection of increased proIL-1 β expression. However, the authors provide evidence suggesting that enhanced Syk and Caspase-8 activity in *Card9*-deficient cells also contributes to enhancement of IL-1 β processing and secretion. Overall, they identify a novel function of *Card9* as a negative regulator of IL-1 β upon bacterial infection of macrophages. The experiments appear to have been performed diligently, with care taken to ensure their reproducibility. I suggest a few experiments below that might improve the study. I have some disagreements on interpretation of the data, but most of those can be resolved by adjusting the wording.

Major

1) Role of Caspase-1 and Caspase-8

The authors state 1) "The mature form of caspase-8 was increased in *Card9*^{-/-} BMDMs at 2 hours (Fig. 2g) suggesting that, in the absence of CARD9, increased caspase-8 activity is responsible for the increased conversion pro-IL-1 β to its mature form." And also 2) "The enhanced production of NLRP3-dependent IL-1 β secretion in *Card9*^{-/-} BMDMs is independent of caspase-1 (Fig. 2g)". I disagree with this interpretation of the data. Even if the cleavage of Casp1 and Casp8 remained the same in Salm-stimulated *Card9* KO (vs wt), an increase in proIL-1 β expression should itself lead to an increase in mature IL-1 β (basic enzyme kinetics - more substrate, more product). In my opinion, the authors should instead write 1) that an increase in Casp8 cleavage may contribute to enhanced IL-1 β processing, and 2) that the enhanced production of NLRP3-dependent IL-1 β secretion in *Card9*^{-/-} BMDMs may involve Casp8. If they want to state that the increase in IL-1 β secretion in *Card9* KOs is truly independent of Casp1, then they should provide experimental evidence for this (e.g. using a Casp1 inhibitor - YVAD or VX-765 - in experiments such as those in Figure 6).

We completely agree that an increase in pro-IL-1 β substrate is likely to lead to elevated levels of mature IL-1 β by simple enzyme kinetics, but in LPS primed macrophages pro-IL-1 β levels are similar between WT and *Card9*^{-/-} (Figure S5 of the revised MS). Upon stimulation/infection of LPS-primed WT and *Card9*^{-/-} cells with nigericin or with *Salmonella* Typhimurium there is an increase in mature IL-1 β in the *Card9*^{-/-} vs WT cells (Figure 6 of the MS) suggesting a considerable effect at the IL-1 β cleavage level (Figure 7 below summarizes the data). We have, however, amended the text as follows to clarify this issue:

“There is, therefore, increased expression of pro-IL-1 β during the “signal 1” phase of inflammasome stimulation in *Card9*^{-/-} cells. However, in WT and *Card9*^{-/-} LPS-primed macrophages, which express similar levels of pro-IL-1 β , the production of IL-1 β is further increased in *Card9*^{-/-} compared to WT cells. This suggests the elevated production of IL-1 β from *Card9*^{-/-} cells is not solely due to the increased pro-IL-1 β substrate availability, but is also due to an effect of CARD9 during “signal 2” of inflammasome activation”.

Figure 7: (a) WT and *Card9*^{-/-} BMDMs express similar levels of pro-IL-1 β after LPS-priming (200 ng/mL for 3 hours). (b-c) LPS-primed *Card9*^{-/-} BMDMs produce more IL-1 β upon infection with *S. Typhimurium* (b) or stimulation with nigericin (c).

2) Is it really deficiency of *Card9* that accounts for the increase in *Salm*-induced proIL-1 β observed in BMDMs? Were the wt and *Card9*-deficient mice used for these experiments derived from homozygous or heterozygous breedings? There have been several papers recently demonstrating that certain passenger mutations acquired during inbreeding can have a frightfully large effect on in vivo and in vitro phenotypes (e.g. mutations in various *DOCK* genes in inbred ASC KO and *NLRP10* KO lines). If het x het breedings were used and the mutation is not genetically linked (i.e. not on the same region of the chromosome as *Card9*) then WT and KO littermates would have an equal chance of inheriting any genetic difference acquired during inbreeding. If WT and *Card9* KO mice were bred independently (especially for many generations), then there is a risk that differences in the genetic

drift/passenger mutations between the WT and KO lines (rather than Card9-deficiency) may account for the phenotype the authors observe. If the authors have used het x het breedings to obtain wt controls, then I am reasonably convinced that the phenotype observed is truly because of Card9 deficiency. If not, the authors have to either confirm their results using littermate controls, or reconstitute Card9 in Card9-deficient cells (e.g. with retroviruses) to show that this normalizes proIL-1b levels and IL-1b secretion. Regardless of how WT controls were produced, this information has to be provided in the Materials and Methods.

We agree with the referee that the number of recent papers suggesting passenger mutations in knock out mice are concerning. We breed from homozygous CARD9 mice and use C57Bl/6 WT to try and manage our animal costs as we have a large colony of different KO mice on a C57/Bl6 background. Our CARD9^{-/-} mice came from David Underhill, who reported these mice had similar phenotypes to data published in CARD9^{-/-} mice generated by other labs see Goodridge, H.S., *et al.*, J. Immunol (2009)¹ and Gross, O., *et al.*, Nature (2006)². We were confident when performing our original work that these mice were genuine knock-outs, but we do appreciate this is not an ideal situation given the increased incidence of passenger mutations appearing in the literature. The time frame to correct the MS is not sufficient for us to breed back to heterozygotes and perform the experiments requested by the referee. We had planned to utilise an alternative approach by reconstituting CARD9 in the Card9^{-/-} immortalised BMDM, but as shown in Figure 5 of this point-by-point response (for reviewer 1 point 4), we find that Card9^{-/-} iBMDM cells have reduced levels of SYK compared to WT iBMDMs. In primary macrophages from WT and Card9^{-/-} mice the levels of Syk are the same and thus the reconstitution experiments were abandoned (see Figure 5 (under the response to referee 1) and Figure S2 of the MS for basal expression in different BMDMs). We have added text to the methods section explaining the status of our mice as requested by the referee:

“Mice were backcrossed on a C57BL/6 background at least 8 generations. All mice strains were bred independently.”

3) Is this a cell-intrinsic phenotype? In other words, does Card9 directly inhibit pathways controlling IL-1b expression and directly inhibit Syk phosphorylation? Supplementary Figure 7 suggests the authors favour this interpretation, presumably on the basis of their immunoprecipitation data which I did not find especially compelling. A possible contribution of autocrine cytokine signalling or other indirect actions of Card9 should not be overlooked. Card9-deficient neutrophils fail to secrete IL-10 in response to Mycobacteria (Dorhoi J Exp Med 2010). The same may be true for macrophages, and lack of autocrine IL-10 signalling could in principle influence pro-IL-1b expression and Nlrp3 activation.

We did not investigate the potential role of autocrine IL-10, but we think it is unlikely to play a major role in our model because the production of IL-1β occurs early in response to cell stimulation (seen by 2h; Figure 1 of the MS), whereas IL-10 is not produced until 4 hours post-cell stimulation^{3,4}. We have amended the manuscript, however, to include the potential for autocrine cytokine production influencing IL-1β production as follows:

“It is possible that the effects of CARD9 on Salmonella-induced inflammasome activity may be indirect, for example by regulating the autocrine production of cytokines such as IL-10 which is reduced in CARD9 deficient neutrophils infected with *Mycobacterium tuberculosis*²⁷. An IL-10-mediated effect of CARD9 on Salmonella-induced IL-1β production from macrophages is unlikely,

however, as elevated IL-1 β production occurs within 2 hours of infection, whereas IL-10 production occurs later in infection²⁸”

4) *The defect in proIL-1b expression is an interesting finding but it requires mechanistic explanation. Card9 is known to promote proIL-1b production in response to Syk-dependent (e.g. CLR ligation) and independent (e.g. Rad50, RIG-I) pathways. What is so different about pathways leading to proIL-1b expression in response to Gram negative bacteria that Card9 plays an opposite role?*

We have thought a lot about the mechanism involved in the response to Gram negative bacterial infection because this is an issue we too have found perplexing. Gram negative bacteria produce many ligands that will activate different PRRs leading to a complex pattern of macrophage stimulation. In our original manuscript we showed an inhibitory role of NOD2 on IL-1 β expression (originally Supplementary Fig. 5, now Supplementary figure 7 of the revised manuscript). This led us to hypothesise that NOD2 activation by bacterial peptidoglycan products would allow NOD2 to cross-talk with the CARD9 pathway we have identified. To test this hypothesis we co-stimulated the cells with a high concentration of MDP to activate NOD2 and, following infection with *Salmonella*, we found decreased pro-IL-1 β expression in WT cells, while no inhibition is observed in *Card9*^{-/-} and *Nod2*^{-/-} BMDMs (Figure 8 below). This cross-talk between CARD9 and NOD2 is likely to be important for controlling pro-IL-1 β expression which could explain the uniqueness of this pathway. We have included this data in the MS (Supplementary Fig. 7d) to support our hypothesis and amended the text to clarify this issue as follows:

“These data are consistent with the NOD2-CARD9 axis negatively regulating pro-IL-1 β production through an action on MAPK signaling independently of NOD2-dependent induction of TNF- α . Consistent with this idea, overstimulation of NOD2 at the same time of infection with *S. Typhimurium* leads to a CARD9 and NOD2-dependent decrease in pro-IL-1 β expression (Supplementary Fig. 7d).”

Figure 8: Cross-talk between NOD2 and CARD9 inhibits pro-IL-1 β expression. Co-stimulation of BMDMs with 10 μ g/mL MDP and *Salmonella* infection leads to decreased pro-IL-1 β expression in WT cells, while no inhibition is observed in *Card9*^{-/-} and *Nod2*^{-/-} BMDMs.

5) *I do not find data in figures 8, S5, or S6 to be compelling evidence that "CARD9 is a central signaling hub for inflammatory signaling". The data is consistent with this proposition, but the proposition itself and the abundant published data to support it are not new. The novelty of this*

study is that it proposes that Card9 can also negatively regulate pro-inflammatory cytokine production. On a similar note I find the title long, uninformative, and not reflective of the central finding of the paper. These decisions I leave to the authors and editors, but a simple alternative would be 'Card9 negatively regulates Nlrp3-induced IL-1 β production upon bacterial infection of macrophages'.

We have changed the title of the MS as follows:

*"Card9 negatively regulates Nlrp3-induced IL-1 β production upon *Salmonella* infection of macrophages"*

Minor

6) In the introduction, the authors state while referring to previous publications that "CARD9 regulates SYK activity and this kinase phosphorylates the CARD domain of ASC when NLRP3, but not NLRC4, is activated to increase IL-1 β and IL-18 production[22-25]". In the CLR signalling pathway, it is thought that Card9 acts downstream of Syk - Syk is phosphorylated at CLRs, Syk phosphorylates PKCs, and PKCs phosphorylate and activate Card9 to trigger recruitment of Bcl10/Malt1 and activation of NF-kB. As far as I am aware, the authors are the first to provide evidence that Card9 can also act upstream of Syk (directly or indirectly), but what they have written above makes it seem like regulation of Syk by Card9 was already known.

We have amended our text as follows in order to make this point clear in the discussion:

"Our data suggests that, in addition to the well described regulatory role for CARD9 downstream of SYK, it is also possible that CARD9 can act upstream of SYK during bacterial infection."

7) The in vitro bacterial counts (Figure S1a-e) should be presented on a linear scale for each MOI.

We have edited Figure S1 as requested by the referee (reproduced below as Figure 9).

Figure 9: CARD9 does not influence bacteria counts *in vitro* and *in vivo*. (a-c) intracellular bacteria counts of WT, *Nlrc4*^{-/-} and *Card9*^{-/-} BMDMs after infection with *S. Typhimurium* SL1344 at MOIs 1, 10 and 50 for 2 (a), 6 (b) and 24 (c) hours. (g-h) Bacteria burden in the spleen (g) and liver (h) after C57BL/6 WT and *Card9*^{-/-} infection with *S. Typhimurium* M525P (4×10^3 CFU). * $p < 0.05$ in comparison to WT (one-way ANOVA with Tukey's multiple comparisons test). (a-e) Data from two independent experiments (mean and s.e.m.).

8) The LDH assay is an assay for lytic cell death, but the authors present the data as viability. It should be briefly indicated in the Materials and Methods how this was calculated, and what was used to determine 0% and 100% viability. How was the potential contribution of bacterial LDH accounted for?

The contribution of bacterial LDH to the total LDH measured is negligible because the conditions used in the assay are unable to lyse bacteria and any LDH from bacteria lysed during the infection is washed away before we measure the total LDH activity. To illustrate this, we have performed the LDH assay comparing LDH activity from live bacteria and BMDM (see Figure 10 below). Despite the high bacterial numbers (ranging from 1000 to 1000000 bacteria per well), the OD was very low compared to that of BMDMs (ranging from 200000 to 12500 cells per well):

Figure 10: LDH production from decreasing numbers of WT BMDM and *S. Typhimurium*

We have included additional information in the Material and Methods to clarify this issue as follows:

“Cytotoxicity was quantified by measuring LDH activity after being released from live cells.

Uninfected BMDMs (ranging from 12.500 to 200.000 cells/well) were used as standards. After treatment, the cells were washed three times with pre-warmed PBS and the intracellular LDH was released by lysing the cells with Triton X-100 1.2% for 1 hour at 37° C. LDH activity was then measured using CytoTox 96 Non-Radioactive Cytotoxicity Assay (Promega)¹⁷. Cellular viability was then calculated in relation to the uninfected control containing 200.000 cells (100% viability).”

9) Glibenclamide can inhibit Nlrp3 activation, but must be used at exceedingly high concentrations that in our hands also inhibit secretion of TNF. I can accept that IL-1b secretion in WT cells is mostly Nlr4-dependent, but the authors have shown (Figure 3f, and Man et al 2014) that Nlrp3 also contributes to IL-1b secretion in response to WT Salm. Therefore, if glibenclamide is truly inhibiting Nlrp3, I find it also somewhat unexpected it has no effect on IL-1b secretion by WT cells (only by Card9 KO). I would suggest that the authors reproduce these results with MCC950/CRID3, since it is a more potent and specific inhibitor of Nlrp3 than glibenclamide.

As requested by the referee we have performed the experiment using MCC950 and added this data to the MS (Figure 4 in the MS, Figure 11 shown below). It shows a similar effect to Glibenclamide and we have amended the text of the MS as follows.

Figure 11: CARD9 selectively negatively regulates NLRP3-induced IL-1 β production. (a-b) Cellular viability and (c-d) IL-1 β secretion from unprimed WT and *Card9*^{-/-} BMDMs infected with *S. Typhimurium* at MOI 10 for 2 (a,c) and 6 (b,d) hours in presence or absence of MCC950. * $p < 0.05$ in comparison to WT (one-way ANOVA with Tukey's multiple comparisons test). (a-f) Data is from three independent experiments (mean and s.e.m.).

"To determine whether the NLR4-independent impact of CARD9 on IL-1 β production resulted from activation of NLRP3 we infected BMDMs with *S. Typhimurium* at an MOI 10 in the presence or absence of the NLRP3 inhibitors glibenclamide or MCC950^{5,6}. At 2 and 6 hours post-infection, both WT and *Card9*^{-/-} BMDMs showed similar levels of cellular viability in the presence or absence of glibenclamide (Fig. 4a-b), while MCC950 slightly inhibited cell death in *Card9*^{-/-} BMDM (Fig. 4e-f). Glibenclamide and MCC950 did not affect IL-1 β production in WT cells, but they reduced the enhancement of IL-1 β production in *Card9*^{-/-} BMDMs to a level comparable to that seen in WT cells. These data suggest that the increased IL-1 β production from *Card9*^{-/-} macrophages after *Salmonella* infection is driven by enhanced NLRP3 activation (Fig. 4c-d,g-h)."

10) Did the authors perform isotype control IPs to determine the (co-)immunoprecipitation is specific? Except for the interaction of ASC with p-Syk, The IP data in Figure 5 is not especially convincing. The pSyk data in *Card9* KO is also not especially compelling.

We have performed isotype control IPs for *Card9*^{-/-} and *Pycard*^{-/-}, and we saw no non-specific immunoprecipitation (see Figure 12 below). We have included the data as a supplementary figure (Supplementary Figure 4) and amended the text of the MS as follows.

"(...)Phosphorylated SYK regulates NLRP3 activation⁷⁻¹⁰, so the interaction of CARD9 with unphosphorylated SYK may prevent the its subsequent phosphorylation thereby inhibiting NLRP3 activation. No proteins were pulled down in *Card9*^{-/-} or *Pycard*^{-/-} isotype control IPs (Supplementary Fig. 4)."

Figure 12: CARD9 and ASC co-IPs isotype controls. *Card9*^{-/-} and *Pycard*^{-/-} BMDMs were primed with LPS (200 ng/mL) for 3 hours and incubated with *S. Typhimurium* (MOI 10, 30 minutes) or with Nigericin (10 μ M, 30 minutes). IPs were then performed as indicated. Images are representative of three independent experiments.

We have repeated the co-IPs and obtained better images (as shown in Figure 4, point 3 of referee 1), which is now included in figure 5b of the revised MS.

11) The authors state "BMDM infection with *S. typhimurium* (MOI 10) in the presence of the caspase-8 inhibitor Z-IETD-FMK, as expected, had no effect on cell viability (Fig. 6a) because caspase-8 does not induce pyroptosis in response to infection with this pathogen." They observed similar results are observed with Nigericin. To say that the caspase-8 inhibitor is not influencing viability (using LDH release after lytic death as a read-out) may not be entirely correct, because it may reduce apoptosis without influencing lytic/pyroptotic death (Sagulenko et al CDD 2013).

The referee is correct we only used the LDH assay to check for cell viability so we could have missed cell death in response to caspase 8 activation driven by other mechanisms. We have added a sentence to the MS as follows to take this comment into account.

"BMDM infection with *S. Typhimurium* (MOI 10) in the presence of the caspase-8 inhibitor Z-IETD-FMK, as expected, had no effect on cell viability as measured by LDH activity (Fig. 6a) because caspase-8 does not induce pyroptosis in response to infection with this pathogen¹⁷. It is possible, however, that caspase-8 may induce cell death in response to infection by other mechanisms³¹."

12) As the authors point out, several GWAS studies have associated hypomorphic alleles of CARD9 with pro-inflammatory diseases, suggesting that CARD9 may negatively regulate inflammation. There is also evidence in mice that Card9 can negatively regulate inflammatory responses. For instance, Card9-deficient mice fail to control Mycobacterium infection, and display an exacerbated (and lethal) inflammatory phenotype (Dorhoi J Ex Med 2010). The failure to effectively control the initial Mycobacterial infection may result from lack of Card9-dependent pro-inflammatory pathways. However, the late hyperinflammatory response may arise from failure of Card9-deficient neutrophils to produce the anti-inflammatory IL-10 (meaning that Card9-deficient neutrophils have a hyperinflammatory phenotype). Similar results are seen upon Listeria infection of Card9-deficient mice (Hsu Nat Immunol 2010). Could the authors' finding that Card9 negatively regulates pro-inflammatory cytokine production be relevant in the context of these previous publications?

We have amended the discussion section of our paper as follows to take this into account.

“Identification of negative regulators of inflammasome activation may have important clinical implications because dysregulated inflammasome activity is associated with a number of important diseases¹¹⁻¹². Genome-wide association studies found strong correlations between loss of function CARD9 mutations and an increased likelihood of developing inflammatory diseases¹³⁻¹⁷. In a *Mycobacterium tuberculosis* infection model CARD9 knockout mice have an increased bacterial burden and develop exacerbated systemic inflammatory responses²⁷, further strengthening the link between CARD9 and inflammatory diseases. Similarly, *Card9*^{-/-} mice are deficient in controlling *Candida albicans* infection, a fungal pathogen capable of stimulating NLRP3⁶. These *in vivo* and clinical observations emphasize CARD9 role as a negative regulator for inflammation, possibly by fine-tuning NLRP3-mediated IL-1 β production”

Reviewer #3 (Remarks to the Author):

Pereira and colleagues demonstrate a novel inhibitory role for CARD9 in IL-1beta production. The authors show that CARD9 reduces SYK mediated NLRP3 inflammasome activation, which in turn leads to a reduction in caspase-8 activity.

The study is well performed and the results appropriately interpreted. This regulatory role for CARD9 in IL-1beta production is both interesting and novel. I do however have a few concerns/comments:

1. Is there biological relevance of this CARD9 mediated regulation of IL-1beta in the setting of salmonella infection in vivo? For salmonella infection the authors show that there is no clear difference in bacterial burdens between WT and Card9-/- (Supp. Fig 1f,g). In addition the increase in pro-IL-1beta appears rather modest (Fig 1p); densitometry of this blot would be helpful.

We saw differences in the bacterial burden between WT and *Card9*^{-/-}, but, disappointingly, they were not substantial enough to reach statistical significance. Differences in bacterial load have been seen in a *Mycobacterium tuberculosis* infection model⁵ supporting our hypothesis that CARD9

contributes to the host response against bacterial infection and we have referenced this paper in our discussion as follows.

“In a *Mycobacterium tuberculosis* infection model CARD9 knockout mice have an increased bacterial burden and develop exacerbated systemic inflammatory responses²⁷, further strengthening the link between CARD9 and inflammatory diseases. Similarly, *Card9*^{-/-} mice are deficient in controlling *Candida albicans* infection, a fungal pathogen capable of stimulating NLRP3⁶. These *in vivo* and clinical observations emphasize CARD9 role as a negative regulator for inflammation, possibly by fine-tuning NLRP3-mediated IL-1 β production”

We have performed densitometric analysis of the blots and added the data in Figure 2 of the revised manuscript, also showed here as figure 1 under the response to Reviewer 1's point 1.

2. The title is somewhat misleading as other NLRP3 agonists seem to mediate a similar response (ie. Nigericin) and hence this is not restricted to bacterial infections. Have the authors examined other NLRP3 agonists to confirm this can be extrapolated to both soluble and crystalline NLRP3 agonists? In addition have the authors examined if the AIM2 inflammasome is affected?

To address this comment we have performed extra experiments using ATP, a NLRP3 agonist, which again showing an increase in IL-1 β production in *Card9*^{-/-} BMDMs (Figure 13 below). We have included the data in the MS (Figure 4).

Figure 13: (a-b) Cellular viability (a) and IL-1 β secretion (b) from LPS-primed BMDMs after ATP stimulation (5 mM, 30 minutes). * p<0.05 in comparison to WT (one-way ANOVA with Tukey's multiple comparisons test).

We have also examined whether there is any role for CARD9 in regulating the activity of the AIM2 inflammasome, by transfecting poly(dA:dT) in LPS-primed WT and *Card9*^{-/-} macrophages. No significant difference was observed. The data is shown below (Figure 14) and is now Supplementary Figure 3 in the MS. The following sentence has been added to the MS:

“To determine whether AIM2 inflammasome activity could be regulated by CARD9 LPS primed WT and *Card9*^{-/-} BMDMs were stimulated with the AIM2 ligand poly(dA:dT), but no differences in cellular viability or IL-1 β secretion were seen (**Supplementary Fig. 3**).”

Figure 14: CARD9 does not control IL-1 β produced via the AIM2 inflammasome. (a) Cellular viability and (b) IL-1 β from LPS-primed BMDMs after transfection with poly(dA:dT) for 4 hours. Data is from three independent experiments (mean and s.e.m.).

We have changed the title of the MS to “CARD9 negatively regulates NLRP3-induced IL-1 β production upon *Salmonella* infection of macrophages”.

3. Can the authors show biological relevance for this pathway utilizing another NLRP3 agonist *in vivo*?

In our work we focussed primarily on *Salmonella* infection. We do not have experience with other *in vivo* models using NLR agonists and so we are not in a position to perform these experiments. *In vivo* data published by other authors suggests a role for CARD9 in infection models known to activate the NLRP3 inflammasome, such as *Candida albicans*² and *Mycobacterium tuberculosis*⁵. We have included a sentence citing these papers in the discussion to support the potential biological relevance of our observations:

“In a *Mycobacterium tuberculosis* infection model CARD9 knockout mice have an increased bacterial burden and develop exacerbated systemic inflammatory responses²⁷, further strengthening the link between CARD9 and inflammatory diseases. Similarly, *Card9*^{-/-} mice are deficient in controlling *Candida albicans* infection, a fungal pathogen capable of stimulating NLRP3⁶. These *in vivo* and clinical observations emphasize CARD9 role as a negative regulator for inflammation, possibly by fine-tuning NLRP3-mediated IL-1 β production”

References

1. Goodridge, H. S. *et al.* Differential use of CARD9 by dectin-1 in macrophages and dendritic cells. *J. Immunol.* **182**, 1146–1154 (2009).
2. Gross, O. *et al.* Card9 controls a non-TLR signalling pathway for innate anti-fungal immunity. *Nature* **442**, 651–6 (2006).
3. Barsig, J. *et al.* Lipopolysaccharide-induced interleukin-10 in mice: role of endogenous tumor necrosis factor-alpha. *Eur. J. Immunol.* **25**, 2888–93 (1995).

4. Foster, G. L. *et al.* Virulent *Salmonella enterica* infections can be exacerbated by concomitant infection of the host with a live attenuated *S. enterica* vaccine via Toll-like receptor 4-dependent interleukin-10 production with the involvement of both TRIF and MyD88. *Immunology* **124**, 469–479 (2008).
5. Dorhoi, A. *et al.* The adaptor molecule CARD9 is essential for tuberculosis control. *J Exp Med* **207**, 777–792 (2010).

REVIEWERS' COMMENTS:

Reviewer #1 (Remarks to the Author):

All my comments have been adequately addressed

Reviewer #2 (Remarks to the Author):

The authors have addressed most of my questions and concerns, and I have no further questions.

Reviewer #3 (Remarks to the Author):

I appreciate the authors responses to my concerns.

I still feel the increase in pro-IL-1beta in Fig. 1p appears rather modest (Fig 1p) and densitometry of this blot would be helpful. The authors have added densitometry - but this was for Fig. 2g.

Once again, we would like to thank the referees for their re-reviews of our manuscript. We have revised our MS to address the remaining issue raised by reviewer 3.

Reviewers' comments:

Reviewer #1:

"All my comments have been adequately addressed"

Reviewer #2:

"The authors have addressed most of my questions and concerns, and I have no further questions."

Reviewer #3:

"I appreciate the authors responses to my concerns. I still feel the increase in pro-IL-1beta in Fig. 1p appears rather modest (Fig 1p) and densitometry of this blot would be helpful. The authors have added densitometry - but this was for Fig. 2g."

We have included densitometric analysis of the immunoblot as requested in Figure 1 of the revised MS.

Figure 1: (a) Immunoblot analysis of pro-IL-1β, caspase-1 and β-actin in spleen cells isolated from infected WT and *Card9*^{-/-} C57BL/6 mice after intravenous infection with *S. Typhimurium* M525P (4×10^3 CFU) and (b) densitometric analysis of this immunoblot.